# Variation of lightning-ignited wildfire patterns under climate change

Francisco J. Pérez-Invernón [1,2] ✉, Francisco J. Gordillo-Vázquez[1], Heidi Huntrieser [2] & Patrick Jöckel [2]

Lightning is the main precursor of natural wildfires and Long-Continuing-Current (LCC) lightning flashes are proposed to be the main igniters of lightning-ignited wildfires (LIW). Previous studies predict a change of the global occurrence rate and spatial pattern of total lightning. Nevertheless, the sensitivity of lightning-ignited wildfire occurrence to climate change is uncertain. Here, we investigate space-based measurements of LCC lightning associated with lightning ignitions and present LCC lightning projections under the Representative Concentration Pathway RCP6.0 for the 2090s by applying a recent LCC lightning parameterization based on the updraft strength in thunderstorms. We find a 41% global increase of the LCC lightning flash rate. Increases are largest in South America, the western coast of North America, Central America, Australia, Southern and Eastern Asia, and Europe, while only regional variations are found in northern polar forests, where fire risk can affect permafrost soil carbon release. These results show that lightning schemes including LCC lightning are needed to project the occurrence of lightning-ignited wildfires under climate change.

Lightning strokes are the main igniters of natural wildfires worldwide. Lightning-Ignited Wildfires (LIW) produce large emissions of carbon, nitrogen oxides and other trace gases[1] playing a key role in climate. The occurrence of lightning-ignited wildfires, in turn, is related to the meteorological conditions that favor the occurrence of lightning and fuel availability. Multiple laboratory experiments [e.g., refs. [2–4]] and field observations [e.g., refs. [5–7]] indicate that continuing electrical currents in lightning flowing for more than some tens of milli-seconds (so called Long-Continuing-Currents, LCC) are likely to produce fires. The evolution and spreading of fires are determined by fuel availability and meteorological conditions, such as air temperature, precipitation rate, and wind strength[8–15].

Projections in the occurrence of lightning over the next century are achieved by using chemistry-climate models and meteorological variables as proxies for lightning occurrence[16–19]. Lightning parameterizations based on the Cloud Top Height (CTH), the cold cloud depth, and the product of the Convective Available Potential Energy (CAPE) and the convective precipitation predict a considerable increase in global total lightning activity, ranging between 5 and 16% at the end of this century[20,21]. On the contrary, parameterizations based on the convective mass flux and the upward cloud ice flux suggest a 15% global decrease on total lightning activity in 2100[22]. Despite global differences in total lightning projections, all the parameterizations suggest a future increase in total lightning activity in Eastern Asia and northern boreal forests driven by warmer air and stronger convection. Investigating the role of climate change associated to the risk of lightning-ignited wildfires in northern boreal forests is essential, as wildfires in permafrost soils can contribute to a significant release of organic carbon to the atmosphere[23].

Krause et al. (2014)[24] investigated the sensitivity of global lightning-ignited wildfires to future variations of lightning frequency by using components of the Max Planck Institute Earth System Model (MPI-ESM) under the Representative Concentration Pathway 8.5 (RCP8.5) scenario. They coupled the European Center HAMburg

[1]Instituto de Astrofísica de Andalucía, Consejo Superior de Investigaciones Científicas, Glorieta de la Astronomía s/n, Granada 18008 Andalucía, Spain. [2]Institut für Physik der Atmosphäre, Deutsche Zentrum für Luft- und Raumfahrt, Münchener Str. 20, Oberpfaffenhofen 82234 Bayern, Germany. ✉e-mail: fjpi@iaa.es

general circulation model 6 (ECHAM6) including a parameterization of lightning[25] with the land surface vegetation model JSBACH and a fire model. The total number of lightning-ignited wildfires was calculated by multiplying the simulated total number of Cloud-to-Ground (CG) lightning flashes by an efficiency factor. They simulated an increase in the total burned area in high-latitude regions and, in turn, a decline in the burned area in South America and Africa driven by a complex relationship between variations in lightning and in fuel availability. Recently, Chen et al. (2021)[23] applied a lightning parameterization based on CAPE and the convective precipitation to the Climate Model Intercomparison Project Phase 5 climate projections under RCP8.5 and hypothesized that lightning increases could induce fire-vegetation feedback in the northern polar forest and a considerable permafrost soil carbon release.

In this study, we use a parameterization of LCC lightning to investigate the sensitivity of global lightning-ignited wildfires to climate change. Prior to the simulation, we confirm the role of continuing current in lightning ignitions by comparing optical lightning data reported by the Geostationary Lightning Mapper (GLM) with forest fire occurrence data provided by the U.S. Department of Agriculture. We use the LCC lightning parameterization implemented by Pérez-Invernón et al. (2022)[26] in the Modular Earth Submodel System (MESSy) for usage within the ECHAM / MESSy Atmospheric Chemistry (EMAC) model. The LCC lightning parameterization is implemented as a complement of the total lightning parameterizations. We simulate the years 2090-2095 under the Representative Concentration Pathway 6.0 (RCP6.0) scenario to estimate the future occurrence of total lightning and LCC lightning and their relationships with the preferential meteorological conditions of lightning-ignited wildfires. We compare the results with a present-day simulation between 2009 and 2011. The simulations show an increase in future LCC lightning occurrence in South America, Central Africa, Australia, Eastern Asia and the western coast of North America and, in turn, no significant variations in future LCC lightning occurrence in northern polar regions. The obtained simulated variation in the occurrence of LCC lightning are connected with changes in the updraft mass flux in thunderstorms that are, in turn, influenced by the temperature profile and the moisture content of the atmosphere in future scenarios.

## Results
### Continuing electrical currents and lightning ignitions
Among 5858 selected lightning-ignited fires, GLM detected a lightning candidate within a 10 km radius around the location of the fire and 14 days before its occurrence for 5574 fires. In turn, 5254 fires were preceded by at least one LCC lightning flash during the 14 days before in the vicinity (up to 10 km) of the ignition point. The mean proximity index[27] of the LCC candidates is 0.56 with 0.24 standard deviation, while 32% of the LCC flash candidates (1703 in total) had a proximity index higher than 0.7. Therefore, we conclude that the percentage of lightning-produced wildfires ignited by LCC lightning ranges between 29% (1703 among 5574) by setting 0.7 as the threshold of the proximity index and 90% (5254 among 5858) by exclusively using the 10 km and 14 days spatio-temporal criterion. Bitzer[28] and Fairman and Bitzer (2022)[29] reported that the ratio of LCC to total flashes within North America throughout the summer is slightly lower than 10%. Therefore, we conclude that the probability of ignition by LCC lightning is larger than the probability of ignition by flashes with no continuing current.

### Lightning projections
The simulation under the RCP6.0 scenario shows an increase of global total lightning and global LCC lightning of $6.4 \times 10^8$ flashes per year and $6.0 \times 10^7$ flashes per year by the 2090s, respectively (Table 1). The simulated globally averaged temperature at the surface increases by about 4 K. Thus, we obtain an increase in total lightning activity of 11% per K, in agreement with the 12% increase in lightning per K reported

by Romps et al. (2014)[20]. The simulated relative increase of the global total lightning flash rate is similar as the relative increase of the global LCC lightning flash rates (43 and 41%, respectively), while the simulated increase of the global cloud-to-ground (CG) lightning flash rate is lower (28%). The p-value calculated by performing a T-test between the projected and the present-day annual averaged of LCC lightning rates is $10^{-4}$, indicating that the obtained differences are statistically significant (see "Methods"). The estimated increase of LCC lightning over land by 47% indicates a higher risk of lightning-ignited wildfires in the future, with a p-value of $3 \times 10^{-4}$ that indicates statistical significance in the difference of the means. Nevertheless, some remarkable differences are obtained for the polar land regions above 60° N degrees latitude, where total lightning increases by 56%, the CG lightning increases by 28% and LCC lightning increases only by 21%, remaining nearly constant given the comparably large interannual variability. The calculated p-value for the projected and the present-day LCC lightning rates in land northern polar regions is 0.38, while the p-value for the projected and the present-day total and CG lightning rates in land northern polar regions are 0.32 and 0.12, respectively. The high values of the p-values indicate that the obtained difference in the means are not statistically significant. These results show that the risk of lightning-ignited wildfires in the northern polar forests may not be as influenced by the future increase of lightning activity as suggested in previous studies[23].

The global occurrence of lightning flashes may increase significantly, especially over land and in the oceanic region of Southeastern Asia (Fig. 1a), in agreement with previous studies[20,22,23]. While in most locations the sign of the change in LCC and CG lightning is similar, there are locations where it may not be (Fig. 1b, c), suggesting that a variation in lightning frequency does not necessarily imply the same variation in the frequency of CG and LCC lightning. In particular, the simulations indicate opposite tendencies of total and LCC lightning in the North and the South of South America, some regions of the Western coast of North America, some regions of central Asia and in the Scandinavian Peninsula, where total lightning decreases, but LCC lightning increases. In the same manner, total lightning increases while LCC lightning decreases in Southern Africa and some regions of the Western coast of Australia. The maximum increases of lightning and LCC lightning are located in Southeastern Asia, where the simulation under the RCP6.0 scenario indicates a large increase of precipitation. There are also some significant differences in the spatial distributions of the variations of total and CG lightning. Total lightning decreases while CG lightning increases in some parts of the North of North America, the Scandinavian Peninsula, Southern Africa, and some parts of Northern Eurasia. In the Western coast of Australia and some regions of Eurasia total lightning increases while CG lightning decreases.

The ratio of LCC to total lightning is parameterized using the updraft mass flux as proxy following Pérez-Invernón et al. (2022)[26] (see "Methods"). Thus, LCC lightning projections are driven not only by the influence of the climate on lightning, but also by the influence of climate on the updraft strength in thunderstorms. The obtained changes in the ratio of LCC to total lightning (Fig. 2a) are connected with variations in the mean updraft mass flux in thunderstorms (Fig. 2b). The criterion to consider the MUMF in the calculation of the annual average is that the cloud thickness is at least 3 km and that the lightning occurrence rate is larger than $1.433 \times 10^{-14}$ m$^{-2}$ s$^{-1}$[16]. The simulated globally averaged mean and standard deviation of the updraft mass flux in thunderstorms during 2011 based on 10 hourly model output instantaneous values are $1.3 \times 10^{-3}$ kg m$^{-2}$ s$^{-1}$ and $1.1 \times 10^{-4}$ kg m$^{-2}$ s$^{-1}$, respectively. However, at Northern polar latitudes the mean and standard deviation are $9.5 \times 10^{-5}$ kg m$^{-2}$ s$^{-1}$ and $1.4 \times 10^{-4}$ kg m$^{-2}$ s$^{-1}$, respectively. This difference of the ratio between the mean and the standard deviation of the updraft mass flux in northern polar and global thunderstorms indicates that the mean updraft mass flux in

**Table 1 | Total lightning, CG lightning and LCC lightning flash occurrence simulated for present day and the RCP6.0 scenario**

|  | 2009–2011 | 2091, 2092, and 2095 | Change |
|---|---|---|---|
| Global Total lightning | 46.6 ± 1.2 fl s⁻¹ | 66.8 ± 1.4 fl s⁻¹ | +43 % (11% per K) |
| Polar Total lightning | 0.16 ± 0.05 fl s⁻¹ | 0.25 ± 0.04 fl s⁻¹ | +56 % (14% per K) |
| Land Total lightning | 23.77 ± 0.4 fl s⁻¹ | 34.92 ± 1.4 fl s⁻¹ | +47 % (12% per K) |
| Polar Land Total lightning | 0.15 ± 0.04 fl s⁻¹ | 0.24 ± 0.04 fl s⁻¹ | +56 % (14% per K) |
| Global CG lightning | 6.1 ± 0.1 fl s⁻¹ | 7.8 ± 0.2 fl s⁻¹ | +28 % (7% per K) |
| Polar CG lightning | 0.06 ± 0.02 fl s⁻¹ | 0.083 ± 0.009 fl s⁻¹ | +38 % (9% per K) |
| Land CG lightning | 3.29 ± 0.05 fl s⁻¹ | 4.3 ± 0.1 fl s⁻¹ | +31% (8% per K) |
| Polar Land CG lightning | 0.06 ± 0.02 fl s⁻¹ | 0.077 ± 0.009 fl s⁻¹ | +28% (7% per K) |
| Global LCC lightning | 2.9 ± 0.1 fl s⁻¹ | 4.1 ± 0.1 fl s⁻¹ | +41% (10% per K) |
| Polar LCC lightning | 0.016 ± 0.004 fl s⁻¹ | 0.019 ± 0.002 fl s⁻¹ | +19% (5% per K) |
| Land LCC lightning | 1.47 ± 0.03 fl s⁻¹ | 2.16 ± 0.07 fl s⁻¹ | +47% (12% per K) |
| Polar Land LCC lightning | 0.015 ± 0.004 fl s⁻¹ | 0.018 ± 0.002 fl s⁻¹ | +21% (5% per K) |

Changes are calculated as the difference between RCP6.0 and the present-day simulations. The errors have been estimated from the standard deviation over the three years that comprise each period. Polar regions only include the North Pole.

thunderstorms is more variable over the North pole than in the rest of the world, possibly due to the smaller sample size. The simulated changes in the mean updraft mass flux (MUMF) are strongly influenced by changes in the total totals index (Fig. 2c). The total totals index is a commonly used stability index calculated as sum of the Vertical Totals Index (temperature at 850 mb minus temperature at 500 mb) and the Cross Totals Index (dew point at 850 mb minus temperature at 500 mb). The simulated variation of the total totals index, in turn, is determined by changes in the vertical profile of the temperature (Fig. 2d) and in the relative humidity (Fig. 2e). The applied parameterization of LCC lightning[26] imposes an opposite relationship between the ratio of LCC to total lightning and the updraft in thunderstorms with high updraft mass fluxes (above ~ 0.2 kg m⁻² s⁻¹). This can be seen in the predicted trend of opposite direction of the mean updraft mass flux and the ratio of LCC to total lightning in some regions, as in South America, the coasts of North America, the Atlantic Ocean, Western Europe and the mainland of Asia. These are regions where the annual averaged mean updraft mass flux in thunderstorms during 2009–2011 is large (Fig. 2f).

**Implications for lightning-ignited wildfires**
While the ignition of lightning-ignited wildfires is influenced by the occurrence of LCC lightning, the survival and arrival phases of fires are determined by the meteorological conditions and the availability of fuel[8]. In particular, the accumulated precipitation during the occurrence of LCC-lightning determines the potential of lightning-ignited wildfires to spread and evolve in a detectable fire[12,30,31]. We simulate the change of the 10-hourly averaged ratio of the LCC lightning density to the total rain rate over land (Fig. 3). We propose using the simulated changes of this ratio as a proxy for lightning-ignited wildfires risk under climate change. Changes in the ratio of LCC lightning to convective rain can serve as an index to determine an increasing occurrence of drier lightning, as dry lightning are associated with lightning-ignited wildfires[9]. High (low) ratios of the LCC lightning density to the total rain rate indicate an increase (decrease) in the occurrence of LCC lightning taking place under lower rainfall rates and, in turn, possible higher (lower) risk of lightning-ignited wildfires in the future. We estimate a decreasing risk of lightning-ignited wildfires in polar regions in the 2090s, with the exception of some small regions distributed over the Scandinavian Peninsula, Alaska, and Siberia, where the risk of lightning-ignited wildfires can be significantly enhanced by the simulated increase of LCC lightning occurrence. In addition, we estimate a pronounced higher risk of lightning-ignited wildfires in Souteastern Asia, South America, Africa, and Australia and a significant change of patterns on a regional scale in North America and Europe. In particular, we estimate a large increase of lightning-ignited wildfires along the

Mediterranean basin and in the Western coast and central part of North America in the 2090s.

The accumulated rainfall, the vapor pressure deficit, the relative humidity and the temperature at 2 m altitude influence the fuel availability and the survival and arrival of fires[8–15]. Thus, we calculate the annually averaged variation of these meteorological variables and the total and LCC lightning in each grid cell under climate change scenarios over the globe (Figure S4). We estimate the annual variation of lightning-ignited wildfire risk in each grid cell as the sum of the percentage annual change of the total lightning, the LCC lightning, the vapor pressure deficit, and the temperature at 2 m altitude minus the percentage annual change of the accumulated rainfall and the relative humidity (Fig. 4). Given a particular grid cell, only variations that are statistically significant are included in the sum. We consider that the risk of lightning-ignited wildfires may increase in cells where the sum of these factors is positive. We obtain that the risk of lightning-ignited wildfire may increase over 77 and 57% of the cells covering the globe and the polar regions, respectively. In the same manner the risk of lightning-ignited wildfire may decrease over 22 and 43% of the cells in the globe and in polar regions, respectively. Importantly, most of the cells presenting lightning-ignited wildfire risk increases in polar regions show a possible increase between only 1% and 5%. Increases in northern polar regions are mainly focused in Greenland, where there is a low fuel availability. On the contrary, there are significant decreases across Siberia and Alaska, two regions where lightning-ignited wildfires are common during the fire season. Therefore, the simulations show a possible tendency to a lower risk of lightning-ignited wildfires in northern polar forest under climate change due to an increase in precipitation in combination with negligible variations in the occurrence of total and LCC lightning.

The obtained variation of LCC lightning is strongly influenced by the lightning parameterization used in the simulations. Nonetheless, the estimated variation of the ratio of LCC to total lightning may be less impacted by the lightning parameterization, as it is calculated similarly for all the grid cells that contain lightning. Therefore, simulated variations in the future ratio of LCC to typical lightning (Fig. 2a) can also serve as a proxy for changes in the risk of lightning-ignited wildfires. As an alternative to estimate the variation in the risk of lightning-ignited wildfires, we estimate the annual variation of lightning-ignited wildfire risk in each grid cell as the sum of the percentage annual change of the ratio of LCC to total lightning, the vapor pressure deficit and the temperature at 2 m altitude minus the percentage annual change of the accumulated rainfall and the relative humidity. We obtain that the risk of lightning-ignited wildfire may increase over 56 and 38% of the cells covering the globe and the polar regions, respectively. In the same

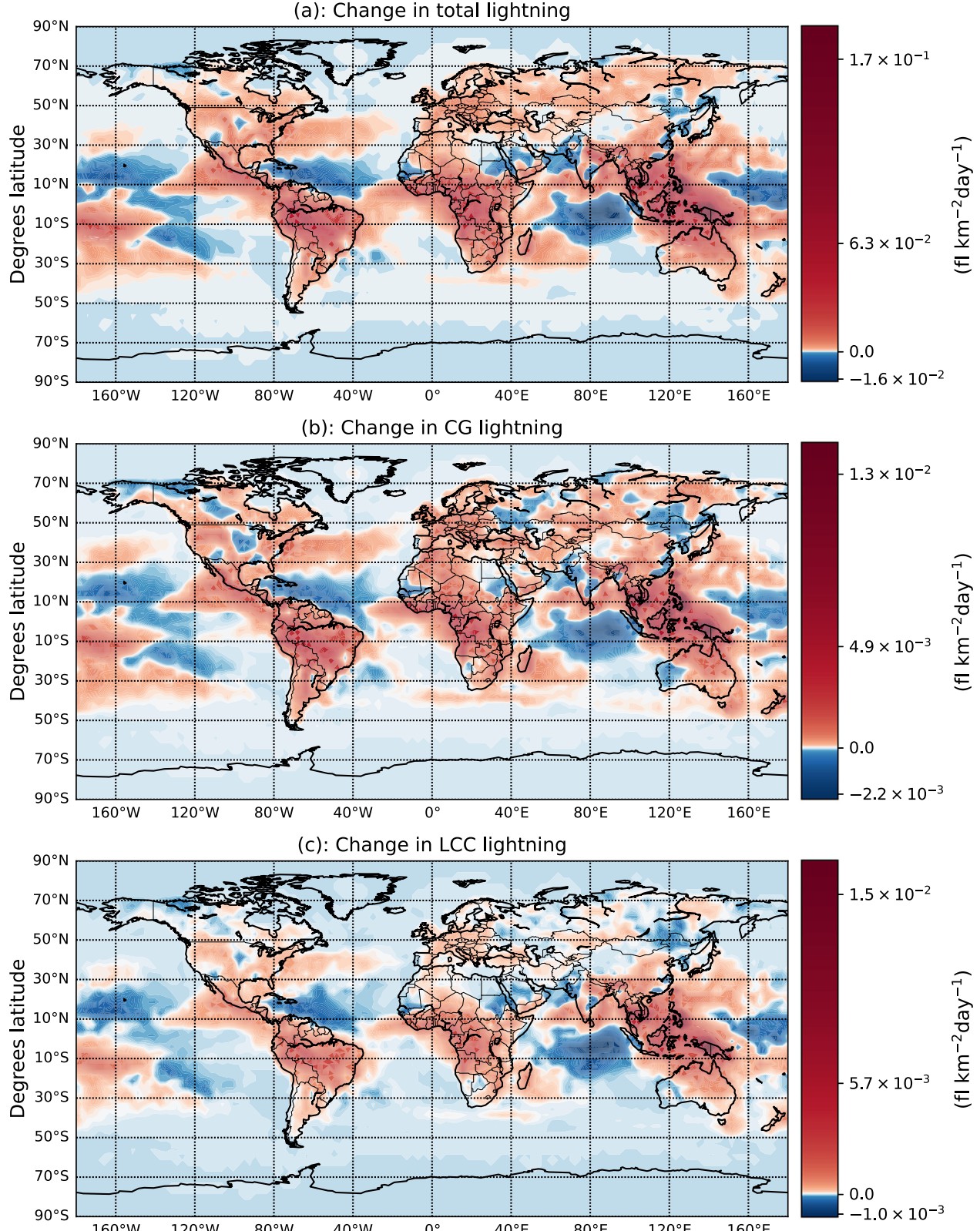

**Fig. 1 | Annually averaged changes of total lightning, Cloud-to-Ground (CG) lightning and Long-Continuing-Current (LCC) lightning.** Annually averaged changes in total lightning (**a**). Cloud-to-Ground (CG) (**b**) and Long-Continuing-Current (LCC) (**c**) lightning flash rate between the periods 2009–2011 and 2091–2095. Changes are calculated as the difference between the Representative Concentration Pathway RCP6.0 and present-day simulations.

manner the risk of lightning-ignited wildfire may decrease over 44 and 62% of the cells in the globe and in polar regions, respectively. The obtained global increase in the risk of lightning-ignited wildfires and the decrease over polar regions following this alternative method provide robustness to our conclusions, despite the uncertainty introduced by the parameterization of total lightning.

The simulated seasonal variations of LCC lightning flashes indicate a possible change in the seasonal occurrence pattern of lightning-

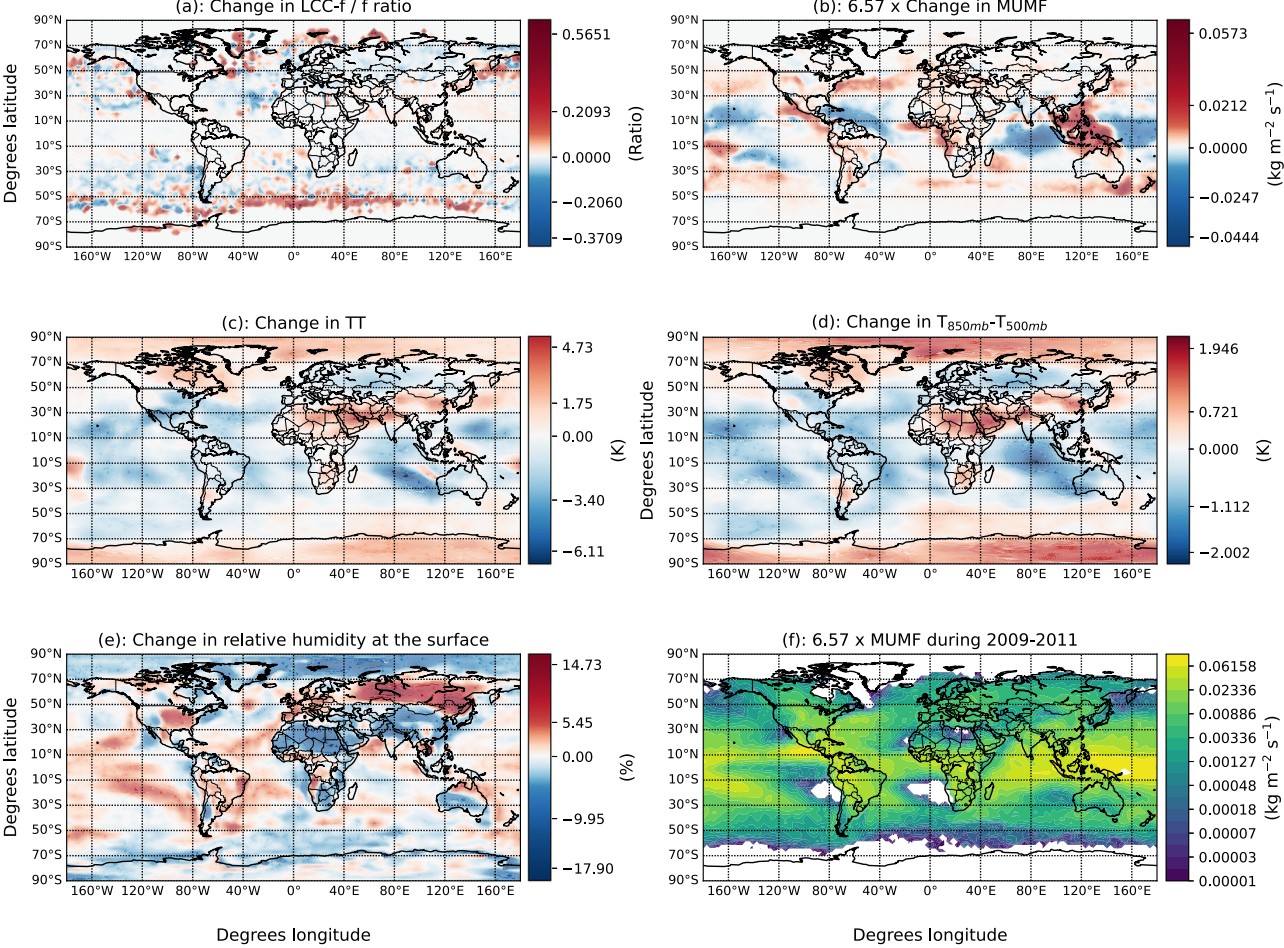

**Fig. 2 | Annually averaged changes of the ratio of meteorological parameters that are relevant for lightning and Long-Continuing-Current (LCC).** The plotted parameters are the ratio of LCC to total lightning (**a**), the mean updraft mass flux (MUMF) at 440 hPa pressure level in thunderstorms (**b**), the total totals (TT) index (**c**), the temperature difference between the 850 mb and the 500 mb pressure levels (**d**), and the change in relative humidity at the surface (**e**) between the periods 2009–2011 and 2091–2095. **f** Annually averaged MUMF at 440 hPa pressure level in thunderstorms during 2009–2011. The criterion to consider the MUMF in the calculation of the annual average is that the cloud thickness is at least 3 km and that the lightning occurrence rate is larger than $1.433 \times 10^{-14}$ m$^{-2}$ s$^{-1}$[16]. The MUMF have been multiplied online by 6.57 as it is applied in the parameterization of LCC lightning[26]. Changes are calculated as the difference between the Representative Concentration Pathway RCP6.0 and present-day simulations.

ignited wildfires under climate change in the 2090s (Fig. S5). The frequency of LCC lightning can significantly increase in some polar areas during summer months (JJA) and, in turn, decrease during the period between March and May (MAM). The months between March and August is the period with the higher risk of lightning-ignited wildfires in northern forests of Eurasia and America[23]. Similarly, increases in the occurrence of LCC lightning in mid-latitudes of the Northern Hemisphere are concentrated between March and August[7,32–34], the period with the highest occurrence of lightning-ignited wildfires in the polar regions. In the Amazon, the most significant increase of LCC lightning is simulated between December and May, while the highest incidence of dry lightning occurs between June and November[35]. The simulations show further that the largest increases of LCC lightning in Australia will occur between December and February, a period with a high occurrence of dry lightning with the potential to produce lightning-ignited wildfires[36]. In Africa, thunderstorms at the end of the dry season are the main precursors of dry lightning[37]. The simulated occurrence of LCC lightning under climate change does not significantly vary in Central and Eastern Africa during the dry season (June through August). In contrast, the future occurrence of LCC lightning is reduced at the end of the dry season in Southern Africa (between September and October). The variation in the risk of lightning-ignited wildfire during the

fire season, when the meteorological conditions favor the arrival and survival phases, is useful to determine the impact of the variation of the risk of ignition in the spreading of wildfires. The simulated variations of the risk of ignition during the fire season (Fig. 5) show a possible increase in the occurrence of lightning-ignited wildfire in Europe, Eastern Asia, North America, the Western coast of South America, central Africa, and Australia. In turn, the simulations suggest a decrease in the risk of lightning-ignited wildfires in polar regions of Eurasia and North America. Finally, projections do not show any clear tendency in the Amazon rainforest during the typical fire season.

## Discussion

Modern results of numerical modeling, remote-sensing techniques, and lightning detection systems have opened the door to a considerable improvement in the knowledge of the relationships between climate, atmospheric electricity, and wildfires in the last decades[20–24,38]. The use of meteorological variables as proxies for lightning occurrence has enabled the implementation of lightning in chemistry-climate models e.g., refs. [16–19], while remote-sensing techniques have contributed to measure the global occurrence of LCC lightning and to develop a parameterization of this type of flash (refs. [7,26]). Simultaneously, the gradual upgrade of lightning detection systems[39] provides

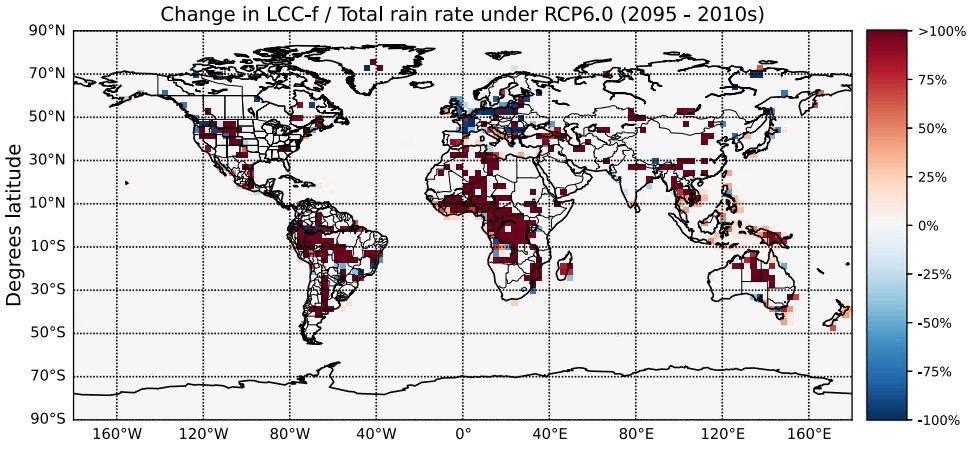

**Fig. 3 | Annual changes (in %) of the 10-hourly averaged ratio between the Long-Continuing-Current (LCC) lightning flash rate and the total rain between 2010 and 2095.** Changes are calculated as the difference between the Representative Concentration Pathway RCP6.0 and present-day simulations in grid cells where the change of LCC lightning rate is statistically significant according to a T-test between the projected and the present-day simulations. The color bar has been deliberately saturated at the upper end due to the high variability of the plotted ratio in some grid cells.

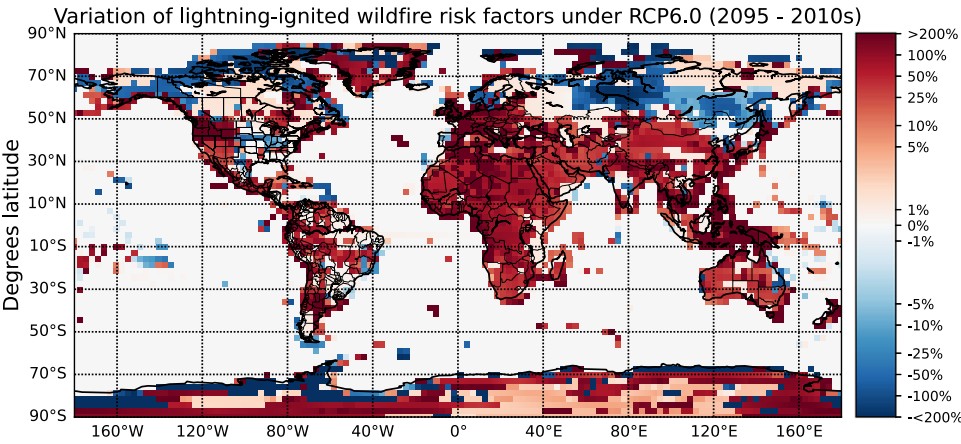

**Fig. 4 | Annually averaged change of the risk of lightning-ignited wildfires in each grid cell under climate change.** Changes in the risk have been calculated from the variation of total lightning, Long-Continuing-Current (LCC) lightning, vapor pressure deficit, temperature, total rain and relative humidity over ocean between the periods 2009-2011 and 2091-2095. The color bar has been deliberately saturated at the upper and the lower ends due to the high variability of the plotted risk.

more confidence in the identification of lightning-igniting flashes and their preferential meteorological conditions[7,11,27,33,34]. These advances may additionally be useful for the improvement of lightning-ignited wildfire prediction with climate models.

Despite important disagreements in lightning projections under climate change scenarios, all lightning projections predict a clear increase of total lightning in polar regions[20–22,24,38]. The projected increase of lightning in northern polar forests could imply an enhancement of the risk of fire and may additionally favor the release of permafrost carbon to the atmosphere[23]. However, current lightning parameterizations do not provide information about the lightning type and polarity, and cannot be used to estimate the risk of ignition by LCC lightning, which could represent a significant portion of the igniters of lightning-produced wildfires according to the presented comparison between lightning flashes recorded by GLM and lightning ignitions provided by the U.S. Department of Agriculture. A recently

developed LCC lightning parameterization[26] has been included in the EMAC model to simulate the sensitivity of lightning and LCC lightning to global change. Globally, we find a 41% increase of LCC lightning that is slightly lower than the projected increase of total lightning. As different lightning parameterizations produce different total lightning trends, a clear conclusion on the future variation of global risk of lightning-ignited wildfires cannot be established. Nevertheless, we have obtained a constant occurrence rate of LCC lightning over polar regions, despite the projected increase of lightning, connected with a projected inhomogeneous change in the mean updraft mass flux in polar thunderstorms. In addition, we have obtained that the ratio of LCC lightning to total lightning will decrease in northern polar regions by the end of the century. In particular, we have found a significant decrease in the ratio of LCC lightning to total lightning over northern polar forest, while the enhancements in polar regions are located in Greenland, which is a region mainly covered by ice without fuel

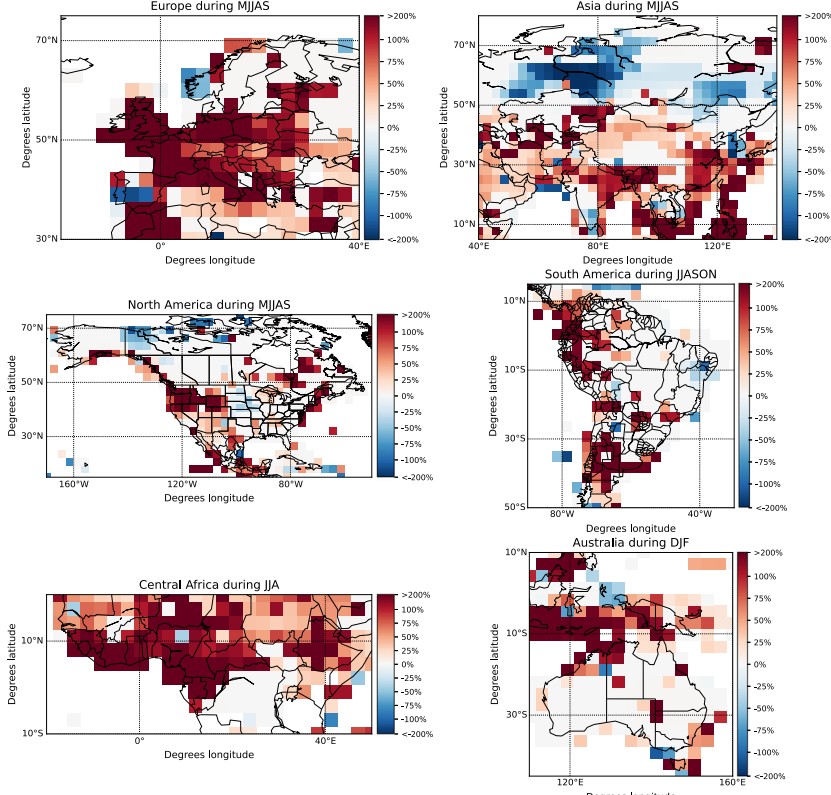

**Fig. 5 | Seasonally averaged change of the risk of lightning-ignited wildfires in each grid cell under climate change over different regions during the lightning-ignited wildfire season.** Changes in the risk have been calculated from the variation of total lightning, Long-Continuing-Current (LCC) lightning, vapor pressure deficit, temperature, total rain, and relative humidity between the periods 2009–2011 and 2091–2095. The color bar has been deliberately saturated at the upper and the lower end due to the high variability of the plotted risk. The seasons are May-June-July-August-September (MJJAS), December-January-February (DJF), June-July-August-September-October-November (JJASON) and June-July-August(JJA).

availability for wildfires. The obtained constancy in the occurrence of LCC lightning in polar regions together with the projected increase in precipitation and relative humidity and the decrease in vapor pressure deficit indicates a possible decrease of the risk of lightning-ignited wildfires in polar regions under climate change. In addition, variations in the simulated global distribution of the ratio of LCC to total lightning suggest a potential change of risk of lightning-ignited wildfires on a regional scale.

These results call attention to the need for including LCC lightning in climate modeling. Further measurements of LCC from optical space-based instruments, such as the Global Lightning Mapper[40] and the Lightning Imager[41] could further enhance the implemented parameterization of LCC lightning. Further research is also needed to identify the preferential meteorological conditions of lightning-ignited wildfires in some regions of the world (as in Africa) where lightning-ignited wildfires are frequent but there are not enough fire reports. Finally, implementation of LCC lightning in climate models coupled with vegetation models could also contribute to improve the estimates of the area burned by lightning-ignited wildfires under climate change.

## Methods

Cloud-to-ground lightning sustaining a long-continuing-current (LCC lightning) have been proposed to be the main ignitors of lightning-ignited wildfires[5–7,42]. The near-field component of the electromagnetic field produced by the continuing phase of LCC lightning decreases with distance following an inverse-cubic law[43,44]. Lightning Location Systems, commonly employed to match fire ignitions and lightning flashes[7,11,33,34,42] rely on very low-frequency (VLF) sensors that are

sensitive to the far-field component of the electromagnetic field[39]. As a consequence, VLF sensors cannot detect the near-field component of distant continuing currents in lightning and, in turn, do not provide information on the role of continuing currents in lightning ignitions. Alternatively, Extreme Low Frequency (ELF) or optical sensors are necessary to investigate fire ignitions produced by LCC lightning. In this study, we have used optical measurements of lightning provided by the Geostationary Lightning Mapper (GLM) aboard the Geostationary Operational Environmental Satellite-16 (GOES-16)[40] combined with a wildland fire database provided by the U.S. Department of Agriculture[45,46] to investigate the role of continuing currents in lightning ignitions.

The GLM provides continuous optical measurements of lightning over the Americas and adjacent oceans since December 2017. The GLM is a Charge-Coupled Device imager with narrow spectral band filters centered on the atomic oxygen line 777.4 nm, usually associated with lightning[47]. The sensor of GLM as well as the data processing are based on the legacy of the low Earth orbit instrument Lightning Imaging Sensor (LIS) mounted on the Tropical Rainfall Measuring Mission (TRMM) between 1998 and 2015[48] and on the International Space Station since March 2017[49].

Adachi et al. (2009)[50] used optical measurements of lightning reported by the Imager of Sprites and Upper Atmospheric Lightning onboard the FORMOSAT-2 satellite to confirm the direct relationship between continuing currents reported by ground-based ELF sensors and optical measurements detected from space. In 2017, Bitzer[28] provided the first global climatology of LCC lightning reported by LIS onboard the TRMM satellite. Bitzer[28] selected flashes detected in five

or more consecutive frames of about 1.9 ms to identify LCC(>9 ms) lightning flashes. A comparison with lightning measurements provided by the Huntsville Alabama Marx Meter Array (HAMMA) allowed Bitzer[28] to show that optically-observed LCC(>9 ms) lightning flashes could have a continuing current lasting more than 22 ms according to the HAMMA system. Recently, Fairman and Bitzer (2022)[29] proposed a new method to classify LCC lightning flashes from GLM measurements based on the identification of the optical characteristics of LCC lightning reported by HAMMA. In this study, we apply the method proposed by Fairman and Bitzer (2022)[29] to classify LCC lightning flashes detected by GLM (Figs. S1 and S2). Only flashes that are not flagged with a degraded quality flag are used in this study. A multiple logistic regression model is applied to each flash by using the GLM attributes maximum distance between two groups, maximum group footprint, maximum group optical energy, maximum number of contiguous groups, median group optical energy and total group optical energy. A flash is classified as a LCC flash if the results of the logistic regression model is larger than 0.7848[29]. We obtain that 10.9% of all the flashes detected by GLM between 15 May 2018 and 31 August 2018 within North America are classified as LCC lightning flashes[51], in agreement with Fairman and Bitzer (2022)[29].

The U.S. Department of Agriculture provides fire data between 1992 and 2018[45,46]. When known, the cause of fire is provided, as well as their coordinates in a 1-square mile resolution grid, date and time. We have extracted the 5,858 lightning-ignited wildfires taking place over Continental United States between 1 June 2018 and 31 August 2018[51] (Fig. S3). We have searched for lightning candidates reported by GLM 14 days before fire occurrence (14 days holdover) and within a 10 km radius around the location of the fire, as previously done by Schultz et al. (2019)[33] by using lightning data from ground-based VLF sensors.

We explore the possible role of LCC flashes in lightning ignitions by searching for LCC lightning candidates among all the lightning candidates reported by GLM and by calculating their proximity index (*A*) according to Larjavaara et al. (2005)[27]:

$$A = \left(1 - \frac{T}{T_{max}}\right) \times \left(1 - \frac{D}{D_{max}}\right), \tag{1}$$

where *D* is the distance between the reported fire location and the lightning flash, *T* is the time between fire ignition and detection, also known as holdover[52], and the parameters $D_{max}$ and $T_{max}$ correspond to the maximum distance and maximum holdover between a fire and a lightning discharge to consider the latter as the potential cause of ignition, respectively. The GLM detects total lightning (CG and IC), while only CG lightning can ignite a fire. Therefore, we cannot assume that the lightning candidate with the maximum proximity index is a reliable candidate. For this reason, we only calculate the proximity index of LCC lightning flashes. Pérez-Invernón et al. (2021)[7] used a fire database over Spain and France where the cause of ignitions were known to establish the typical proximity index of lightning candidates. They estimated the threshold value of the proximity index that ensures that at least 80% of the selected fires are ignited by lightning when comparing the total fire and the lightning databases. They found that the threshold does not significantly depend on the lightning location system. In turn, they reported a threshold of 0.7 in the proximity index value that ensures that at least 80% of the selected fires are ignited by lightning when using $D_{max}$ = 10 km and $T_{max}$ = 14 days. Following this approach, we establish two different criteria to determine that a lightning-produced wildfire was ignited by a LCC flash. The first criterion is less conservative and can enable us to determine an upper limit on the percentage of lightning-produced wildfires ignited by LCC flashes, while the second criterion is more conservative and can be used to obtain a lower limit:

1. Less conservative criterion: The GLM reported at least one LCC lightning flash candidate within 10 km around the fire and 14 days before the fire date.
2. Conservative criterion: The GLM reported at least one LCC lightning flash candidate with a proximity index greater than 0.7.

The EMAC model is a numerical chemistry-climate model that couples the fifth generation European Center HAMburg general circulation model (ECHAM5;[53]) and the second version of Modular Earth Submodel System (MESSy) to link multi-institutional computer codes, known as MESSy submodels[54,55]. Such submodels are used to describe tropospheric and middle atmosphere processes and their interaction with oceans, land, and influences coming from anthropogenic emissions.

The model is run at the T42L90MA resolution, i.e., with a 2.8° × 2.8° quadratic Gaussian grid in latitude and longitude with 90 vertical levels reaching up to the 0.01 hPa pressure level and with 720 s time step length[55]. We use the Tiedtke convection scheme[56] implemented in the submodel CONVECT.

Two simulations are performed covering the years 2009-2011 (present-day) and the years 2090–2095 under the Representative Concentration Pathway 6.0 (RCP6.0) scenario. For the 2009–2011 simulation, we employ the namelist setup for purely dynamical simulations (referred to the E5 setup, no chemistry) in the mode of free running simulation. We start the simulation on January, 2009 using ERA-Interim reanalysis meteorological fields[57] as initial conditions. For the RCP6.0 simulations, we start the simulation of January 2090 and run one year of spin-up to reach an equilibrium. The LCC and total lightning flash density, the total and the convective rain at the surface, the mean updraft mass flux at 440 hPa pressure level and the vertical profiles of the relative humidity and the temperature are output every month in 2091, 2092, and 2095 and every 10 h between 2009 and 2011 and in 2095. We choose 10-hourly output because 24/10 is not an integer in order to avoid systematic local biases with regard to, for example, the solar zenith angle. Monthly averaged outputs provide information about the sensitivity of LCC and total lightning to climate change, while 10 hourly averaged meteorological parameters are useful to estimate potential changes in the occurrence of dry lightning, the main precursors of lightning-ignited wildfires[7,9,12,13,30,31]. The interannual variability of the simulation is used to quantify the difference between the arithmetic means of the projected and the present-day LCC lightning rates by performing a T-test to calculate the corresponding *p*-value[58]. The *p*-value is the level of marginal significance within the statistical hypothesis of equal mean values, indicating the probability of an equal mean for both samples. If the p value is lower than 0.05 (less than 5% probability of equal mean), we reject the hypothesis of equal means and consider that projected and present-day LCC lightning rates are statistically different.

The RCP6.0 simulation is set-up following the simulation RC2-base-04 of Jöckel et al. (2016)[55]. The sea surface temperatures (SSTs) and the sea-ice concentrations (SICs) are prescribed from simulations with the Hadley Center Global Environment Model version 2 - Earth System (HadGEM2-ES) Model[59,60]. Projected mixing ratios of the greenhouse gases and $SF_6$ are incorporated from Eyring et al. (2013)[61]. Anthrophogenic emissions are taken from monthly values provided by Fujino et al. (2006)[62] for the RCP6.0 scenario. We refer to Jöckel et al. (2016)[55] for more details about the simulation set-up.

The LNOX submodel of MESSy estimates the total lightning flash density, the LCC lightning flash density, and the production of $NO_x$ by lightning by using different lightning parameterizations[16] and a scaling factor that ensures a global lightning occurrence rate of ~45 flashes per second[48,63]. For the present study, we use the lightning parameterization based on the Cloud Top Height (CTH)[25] for land and on the convective precipitation[64] for ocean as proposed by Pérez-Invernón et al. (2022)[26] and scaled offline by a factor of 1.13. This

lightning parameterization overestimates the lightning flash density over the ocean with respect to land, producing a land/ocean contrast of about 1:1 instead of the 3:1 contrast reported by space-based instruments[48,49,63]. Nevertheless, this lightning parameterization produces a good agreement between the simulated and the observed spatial distribution of lightning over land[26]. The ratio of cloud-to-ground to intra-cloud lightning is calculated from the cold cloud thickness according to Price and Rind (1993)[65]. The LNOX submodel calculates the LCC(>9 ms) and the LCC(>18 ms) lightning flash frequencies from the mean updraft mass flux multiplied by 6.57 online and by using the parameterizations of the ratio of LCC to total lightning developed by Pérez-Invernón et al. (2022)[26]. The parameterization of LCC lightning by Pérez-Invernón et al. (2022)[26] is non-monotonic in the convective mass flux. Non-monotonic relationships between atmospheric electricity, meteorological parameters[66,67], and aerosol concentrations are common in literature[68–71]. In the case of LCC lightning, Bitzer (2017)[28] proposed a possible link between the occurrence of LCC lightning and the convective mass flux. Lapierre et al. (2014, 2017)[72,73] showed that the physical mechanisms producing continuing current in lightning are possibly different for positive and negative CG lightning. Lapierre et al. (2017)[73] reported that in-cloud negative leaders of positive cloud-to-ground lightning can inject high amounts of charges in the return stroke, producing a long continuing current. However, Lapierre et al. (2014)[72] did not find any clear relationship between the propagation of in-cloud positive leaders and negative cloud-to-ground lightning. These results suggest that the structure of electrical charges in thunderclouds could have multiple effects in the occurrence of continuing currents. On the one hand, the structure of the electrical charged layers plays an important role in the polarity of cloud-to-ground lightning [e.g., Rust et al. (2005)[74]]. In the same manner, the structure of electrical charged layers in thunderstorms is closely related with the meteorological conditions[75]. Therefore, we can expect a non-linear relationship between the meteorological conditions of thunderstorms and continuing currents. On the other hand, Lapierre et al. (2017)[73] reported that the propagation of the in-cloud negative leader supplies the continuing current of positive cloud-to-ground lightning. The propagation of the negative leader is influenced by the distribution of electrical charges in thunderstorm, introducing complexity to the mechanism behind the occurrence of the continuing current.

Bitzer (2017)[28] compared the duration of the optical signal of lightning flashes reported by the Lightning Imaging Sensor (LIS) on-board the Tropical Rainfall Measuring Mission (TRMM) satellite with the duration of the continuing current reported by the Huntsville Alabama Marx Meter Array (HAMMA). He found that a flash with an optical duration of 7–9 ms would have a continuing current lasting 22 ms that is, in turn, in agreement with the minimum duration of fire-igniting flashes reported by Mceachron and Hagenguth (1942)[2] and by Fuquay et al. (1967)[5]. As explained in "Continuing electrical currents and lightning ignitions", we focus on LCC(>9 ms) lightning flashes because they are frequently associated with lightning ignitions. The simulated ratio of LCC(>9 ms)-lightning to total lightning flashes at a global scale is about $6.2 \times 10^{-2}$, while ISS-LIS reported a ratio of $6.6 \times 10^{-2}$[26]. Pérez-Invernón et al. (2022)[26] calculated the seasonal spatial correlation coefficient between the simulated ratio of LCC(>9 ms) to total lightning and the ratio reported by ISS-LIS, obtaining that it increases from nearly 0.2 in DJF to about 0.5 in JJA.

## Data availability

The data of the simulations generated in this study have been deposited in the Zenodo repository [https://doi.org/10.5281/zenodo.6627112 (last access 04-01-2023)]. The processed LCC lightning flashes reported by GLM between 15 May and 31 August 2018 and the processed lightning-ignited wildfires data are available at Zenodo [https://doi.org/10.5281/zenodo.7503122 (last access 04-01-2023)]. The GLM lightning data sets may be obtained from NOAA via their CLASS service [https://www.avl.class.noaa.gov/saa/products/search?datatype_family=GRGLMPROD, last access 08-06-2022][76]). The data of lightning-ignited wildfires used in this study are freely available the USDA website [https://www.fs.usda.gov/rds/archive/Catalog/RDS-2013-0009.5 (last access 08-06-2022)][45,46].

## Code availability

The scripts used for the identification of LCC lightning flashes as lightning-ignited wildfires candidates are freely available under (https://doi.org/10.5281/zenodo.7318114, last access 14-11-2022[51]). The Modular Earth Submodel System (MESSy) is continuously developed and applied by a consortium of institutions. The usage of MESSy and access to the source code is licensed to all affiliates of institutions which are members of the MESSy Consortium. Institutions can become a member of the MESSy Consortium by signing the MESSy Memorandum of Understanding. More information can be found on the MESSy Consortium website (http://www.messy-interface.org, last access: 05-10-2021). As the MESSy code is only available under license, the code cannot be made publicly available. The parameterization of LCC-lightning has been developed based on MESSy version 2.54 and is included in version 2.55.

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

## Acknowledgements

The authors would like to thank NOAA for providing GLM lightning data, the National Interagency Fire Center for providing lightning-ignited wildfire data and the ECMWF for providing the data of ERA-Interim forecasting models. The EMAC simulations have been performed at the German Climate Computing Center (DKRZ) through support from the Bundesministerium für Bildung und Forschung (BMBF). D.K.R.Z. and its scientific steering committee are gratefully acknowledged for providing the HPC and data archiving resources.

F.J.P.I. acknowledges the sponsorship provided by the Federal Ministry for Education and Research of Germany through the Alexander von Humboldt Foundation and the sponsorship provided by Junta de Andalucía under grant number POSTDOC-21-0005 (F.J.P.I.). The project that gave rise to these results received the support of a fellowship from "la Caixa" Foundation (ID 100010434). The fellowship code is LCF/BQ/PI22/11910026 (F.J.P.I.). Additionally, this work was supported by the Spanish Ministry of Science and Innovation, under projects PID2019-109269RB-C43 (F.J.G.V.) and FEDER program. F.J.P.I. and F.J.G.V. acknowledge financial support from the grant CEX2021-001131-S funded by MCIN/AEI/ 10.13039/501100011033.

## Author contributions

F.J.P.I. conceived and designed the experiments, performed the experiments, analyzed the data, contributed materials/analysis tools, and wrote the paper. F.J.G.V. and H.H. analyzed the data and wrote the paper. P.J. analyzed the data, contributed materials/analysis tools, and wrote the paper.

## Competing interests

The authors declare no competing interests.
