## [Peer Review File · Nature Communications]

Variation of lightning-ignited wildfire patterns under climate changeREVIEWER COMMENTS

Reviewer #1 (Remarks to the Author):

Recommendation: Accept Pending Revisions

General Comments

The manuscript is well written and investigates an interesting and potentially important topic of varying patterns of lightning-ignited wildfires under climate change. I believe this is novel analysis and with some revisions would be a solid peer-review contribution to the field of study.

I think the paper would be improved if there were a figure and discussion that shows where lightning, LCC, temperature, and vapor pressure deficit increase and relative humidity and precipitation decrease. I think showing areas where this occurs and where the opposite occurs would bolster your analysis on the varying patterns of lightning-ignited wildfires. This would be a more comprehensive analysis and illustrate with less caveats the modeled changes. Additionally, focusing more on the wildfire aspect rather than changes in lightning would help.

Related to this, I believe there are assumptions stacked on each other to reach certain conclusions without appropriate caveats or qualifiers. First, changes in LCC and lightning are modeled (and they are some assumptions based on that modeling). Then authors emphasize the LCC-wildfire relationship, which is not that robust. Finally, the authors indicate lightning-ignited wildfires will increase in areas where convective rainfall and RH decrease with steady or increasing LCC, respectively. While there is some peer reviewed literature that supports this logical framework, I think the authors should use more caveats or add more context when showing results and especially in the discussion. The findings are valid, but I think being more transparent about some of the less robust relationships and assumptions in the logical framework to get to those findings is warranted.

Finally, regarding just the modeled lightning and LCC determination, any CG lightning pulse can start a fire. Much of it depends on fuel moisture, near-term fire weather, and if lightning hits fuel (i.e., not rocks). I am not asking for the authors to remove the LCC portion of the analysis, but I think recognizing it is a relatively new way of determining LCC and that the LCC-wildfire relationship is not the most robust would strengthen the manuscript. Additionally, I do have some issue that no CG lightning was utilized in this research. Relying on GLM, which cannot distinguish between IC and CG and have resolution and detection problems, could be a major sticking point for some readers and reviewers. I liked how the two approaches were explained, and I think that is sufficient for me. However, this point also supports my recommendation for couching your results and conclusions with more caveats and perspective, while strengthening your analysis with a few additional items and discussion.

Specific Comments

- Lines 101-115: Why wasn't any cloud-to-ground lightning flash/stroke data used to supplement or augment this analysis? You cannot definitively declare that GLM detected lightning was IC or CG.
- Lines 107-109: I don't think you can conclude anything about proportions of LCCs starting fires from these results. Not all LCCs are CG as well.
- Line 119: Table 1, why 2091-92 and 2095 and not 2091-2095?
- Lines 146-149: Interesting result
- Lines 152-185: This paragraph is a bit dense and tough to glean what the authors want to convey
- Lines 163-166: Is this due to smaller sample size of thunderstorms and fewer variations on how and the types of thunderstorms develop?
- Lines 166-170: Not quite understanding this statement. Why would an overall quadratic relationship lead to varying ratios, including perhaps regionally to hemisphere-wide? Trying to articulate changes or diverges in relationships between LCC and updraft flux?
- Lines 172-175: What about high elevation thunderstorms where Totals Totals doesn't capture the convective potential well?
- Line 197: Maybe change dry to "drier"? "Dry" thunderstorms are somewhat regionally defined. If lightning to precipitation ratio changes, it doesn't necessarily mean it is "dry" as it may not reach those more regionally defined thresholds.
- Lines 217 and 210-240: Change "will" to "may". This refers to my general comments where they are some assumptions stacking on assumptions here. Assuming that this method can't accurately model increases in LCC, then assuming that LCCs have a robust relationship with starting fires or a minimum more efficient than other CGs, then assuming a decrease in relative humidity would lead to an increase in lightning-ignited wildfires. I am not saying these are poor methods or the results are invalid, but I am cautioning the authors to use caution when declaring relationships or conclusions based on this research without proper caveats or qualifiers.
- Figure 4: I would like to see more figures that show the potential changes in lightning-ignited wildfire patterns and frequencies. Are you able to produce a plot that shows where lightning and LCC frequency increases with lowered RH, higher temperatures, and possibly increased vapor pressure deficit? Using LCC with change in RH is a good start, but to improve the manuscript and better show how lightning-ignited wildfire patterns will change, I think another figure illustrating all those changes that lead to a likely increase in lightning-ignited wildfires would be helpful.
- Lines 288-290: Why is there a 66% increase? Could you go into this more?
- Lines 307-312: I think there are other factors that are more important to lightning-ignited wildfires than LCC. I think changes in LCC should be a supplemental piece to the change, rather than the primary driver of your results and conclusions on lightning-ignited wildfire pattern changes.
- Lines 366-395: You do a good job of explaining some of the caveats and walking through your methods here. I think you just need to carry some of the caveats over to the results and discussion sections by changing wording (i.e., will to may, responsible to associated) and providing more context.

- Lines 409-425: Why did you choose 2091, 2092, and 2095 and 2009-11 as your periods?

Reviewer #2 (Remarks to the Author):

This manuscript applies the first author's parameterization for long-continuing-current (LCC) lightning flashes to estimate the future change in LCC with global warming.

I do not trust the lightning parameterizations used here, so I do not have much faith in the results. The study parameterizes total lightning as a function of cloud-top height over land using Price and Rind (1992) even though better schemes are available. Then, LCC is parameterized as a fraction of that total lightning, with the fraction given by a strange quadratic function published in the first author's 2022 paper in Geoscientific Model Development (GMD). I say "strange" because the LCC fraction is non-monotonic in the convective mass flux, coming back down to zero and then to negative values for sufficiently intense storms. The GMD paper gives no explanation or justification for this behavior. I tend to believe that the non-monotonic behavior is telling us that LCC should not be parameterized in terms of 2.5x2.5-degree mass flux. On the other hand, if the reported increase in fire is coming not from the quadratic dependence of LCC on mass flux, but from increases in the cloud-top height from the Price and Rind scheme for total lightning, then these results are not novel (see, e.g., Krause et al, 2004).

Reviewer #3 (Remarks to the Author):

In this study, the authors begin by arguing using observations that long continuous current (LCC) lightning could account for 12 to 68% of lightning-ignited fires in the United States. They argue that since LCC flashes account for less than 10% of total lightning flashes, this suggests that LCC flashes are more effective than regular flashes at starting fires, meaning that any changes in LCC lightning in particular might be something we want to pay close attention to under climate change. To investigate that further, they then apply a recently developed parameterization for the fraction of LCC lightning, based on updraft mass flux, in simulations in present-day and future climates using the ECHAM5 model. For a sense of the total flash rate over land in each simulation, a traditional cloud-top-height-based parameterization is used.

Their primary finding is that fire risk may decrease with warming in the Boreal regions due to an increase in precipitation and little-to-no significant change in the LCC lightning flash density. This is something that would not have been possible to determine by looking at changes in total lightning flash density alone, and runs in contrast to the findings of Krause et al. (2014). In other regions, LCC lightning flash density tends to increase, and this is often accompanied by an increase in precipitation making changes in overall lightning-ignited fire risk less clear. There is understandably a fair amount of uncertainty associated with these results, however, which I think the authors do a reasonable job of communicating, but there are places where I think they could do better, which I cite in my major comments.

In general I agree with their conclusion that the contrasting results with Krause et al. (2014), who used a more comprehensive setup with an atmosphere model coupled to a land surface model that included fires, motivate a followup study in which LCC lightning, and in particular the connection between LCC lightning and fire ignition, are included in a similar Earth system modeling framework. While the authors work hard to approximately capture changes in lightning-ignited fire risk using physically motivated logic, these estimates would be more robust with a model that helped explicitly estimate changes in area burned. More years of simulation in the present-day and future climates would also contribute to increased confidence in the conclusion. Overall though, with some improvements, the publication of this manuscript could serve as a good precursor for that followup work.

Major comments

- While for some area-averaged regional change metrics a t-test is used to assess statistical significance (e.g. in the first paragraph of Section 2.2), no test appears to be applied to changes in

individual gridpoints on maps. I think it would make the results of this study easier to interpret if a similar t-test were applied to individual gridpoints, and then a mask (or hatching) were applied to all gridpoints where the change was not statistically significant in the maps. I believe this would be useful since in the text a number of important conclusions are made regarding specific (fairly local) regions, and it would be good to be able to confirm whether these changes are statistically significant. Something similar could be said for the scatterplots in Figure 4. Some gridpoint-level changes in LCC lightning flash density in particular are quite small and seem they may not be outside the range of interannual variability. Should they be included in the percent of points with decreased or increased LIW risk?

- Are there other possible choices for an LCC fraction parameterization? Could that introduce uncertainty? How much do we trust the existing scheme, whose details--importantly the threshold at which the LCC ratio begins to decrease with increasing updraft mass flux--may depend on the circulation model that was used to generate the updraft convective mass fluxes to fit it? Is the uniform scale factor applied to the updraft mass fluxes before passing them into the scheme enough to address this? Since the LCC scheme is fairly central to the conclusions of this study, including some discussion of this feels important.

Specific comments

Lines 48-53: were these studies all using the same emissions scenarios? That seems like something important to control for; otherwise it might be better to report % increase or decrease per degree warming.

Line 64: "atmospheric model European Center HAMburg general circulation model 6 (ECHAM6)" having "atmospheric model" and "general circulation model" in such close succession is a bit awkward to read and feels somewhat redundant. Could one of those descriptors be dropped or are they part of the official name?

Lines 79-82: does focusing on the relationship between LCC lightning and fires only in the United States bias these results at all? Could looking at different land and fire regions (e.g. with different vegetation types) change these conclusions?

Lines 101-115: it is understandable that measuring the causal relationship between lightning and fires is difficult from satellite data alone. Thank you for acknowledging the large uncertainty in these statistics.

Lines 131-134: I just want to confirm that this is over land as well (if it is, maybe refer to this as "polar land regions"). If not, it seems that it would be more relevant if it were focused on land, since it is used to comment on the risk of lightning-ignited fires in polar forests. More generally, while I think it is fair to include the global total lightning change statistics to compare to previous literature, the focus of this paper should be over land, since that is where fires occur.

Table 1: for completeness it might be good to include the total land lightning flash rate statistics (to compare with the LCC land lightning statistics).

Lines 137-139: I suppose since some fires are started by non-LCC lightning that an increase in total lightning may still result in an increase in lightning-ignited fires (even if LCC lightning is effectively constant). Maybe phrase this as "[...] may not be as influenced by the future increase of lightning activity as suggested in previous studies."

Lines 142-145: to my eye, particularly over land the spatial pattern of the change in total and LCC lightning looks to be quite similar to first order. There are indeed exceptions, but they are difficult to pick out. Perhaps adding a panel that showed a map of where the change was statistically significant in both the total and LCC cases, but of opposite sign would help the reader confirm what is written in the text a little more quickly.

Lines 145-149: in looking at the figure in most of these regions it looks more like total lightning decreases and LCC lightning increases. Is this statement correct?

Lines 158-159: how is it determined what a "thunderstorm" is in this context? Is it whether the total lightning parameterization predicts non-zero lightning?

Figure 2: the colormaps in (a) and (b) are very difficult to read near zero -- they seem to perhaps make small changes appear larger than they really are. Is there a way to address this?

Figure 2(a-e): I recommend using symmetric colorbars centered at zero for these change plots. It generally makes them easier to read and interpret.

Figure 2(a-e): this is a bit of a style nit-pick, but more generally, why are the colorbar tick labels unevenly spaced with different formats? With the exception of panel (f), I think there is no need to use scientific notation (e.g. " 5.7×10^{-1} ") is more clearly represented as "0.57"; " 1.5×10^1 " is more clearly represented as "15" etc.).

Lines 192-193: does "dry" lightning ever occur in these simulations? In that case this ratio would be unboundedly large. Also, why restrict to using the convective rain rate in the denominator? Would total rain rate not make more sense here?

Line 205: typo "Southeastern"

Figure 3: the colorbar is saturated in most regions of the world here. I would suggest extending the range. Perhaps more generally though, would it make more sense to pose this as a percent change? I don't have a very good handle on the order of magnitude of this ratio in the absolute sense, and I suspect other readers do not either, so it is hard to determine whether these changes are substantial deviations from the climatology or not.

Line 218: does "the globe" include or exclude the polar regions? Would it be more effective to use non-overlapping regions?

Figure 4: can you confirm whether these changes are considered only over land? I assume they are, but I just want to make sure.

Lines 419-420: is there a reason for excluding years 2093 and 2094 from the analysis? Is it just that the output did not exist?

Line 459: where does the factor of 6.57 come from? I think you mention it briefly in the caption of Figure 2, but a more detailed explanation could be helpful (i.e. acknowledging that it is needed to scale the the updraft mass fluxes in ECHAM such that they approximately match the magnitude of the updraft mass fluxes in ERA5, the dataset the parameterization was derived with).

Response to reviewers

We appreciate the time and effort that the reviewers dedicated to providing feedback on our manuscript and are grateful for the insightful comments and valuable improvements to our paper. We have incorporated most of the suggestions made by the reviewers. Those changes are highlighted within the manuscript. Please see below, in blue, for a point-by-point response to the reviewers' comments and concerns.

Reviewer #1 (Remarks to the Author):

Recommendation: Accept Pending Revisions

General Comments

The manuscript is well written and investigates an interesting and potentially important topic of varying patterns of lightning-ignited wildfires under climate change. I believe this is novel analysis and with some revisions would be a solid peer-review contribution to the field of study.

I think the paper would be improved if there were a figure and discussion that shows where lightning, LCC, temperature, and vapor pressure deficit increase and relative humidity and precipitation decrease. I think showing areas where this occurs and where the opposite occurs would bolster your analysis on the varying patterns of lightning-ignited wildfires. This would be a more comprehensive analysis and illustrate with less caveats the modeled changes.

Following this suggestion, **we have replaced the analysis of the ratio of LCC lightning to relative humidity with two figures** showing a clearer analysis on the varying patterns of lightning-ignited wildfire. In the new Figure 4, we show the annually averaged change of the risk of lightning-ignited wildfires in each grid cell based on the variation of total lightning, LCC lightning, total rain, vapor pressure deficit, relative humidity and temperature. In addition, we have added a new figure to the supplement (Figure S1) showing the annually averaged change of each of these variables.

Additionally, focusing more on the wildfire aspect rather than changes in lightning would help. Related to this, I believe there are assumptions stacked on each other to reach certain conclusions without appropriate caveats or qualifiers. First, changes in LCC and lightning are modeled (and they are some assumptions based on that modeling). Then authors emphasize the LCC-wildfire relationship, which is not that robust. Finally, the authors indicate lightning-ignited wildfires will increase in areas where convective rainfall and RH decrease with steady or increasing LCC, respectively. While there is some peer reviewed literature that supports this logical framework, I think the authors should use more caveats or add more context when showing results and especially in the discussion.

We think the new Figures 4 and S1 provide a deeper focus on the wildfire aspect now. In addition, **we have now moved the seasonal variation of LCC lightning to the supplement and introduced a new figure** (Fig. 5) showing the variation of the risk of lightning-ignited wildfire in different regions of the world during the fire season in order to focus even more on the wildfire aspect.

The findings are valid, but I think being more transparent about some of the less robust relationships and assumptions in the logical framework to get to those findings is warranted. Finally, regarding just the modeled lightning and LCC determination, any CG lightning pulse can start a fire. Much of it depends on fuel moisture, near-term fire weather, and if lightning hits fuel (i.e., not rocks). I am not asking for the authors to remove the LCC portion of the analysis, but I think recognizing it is a relatively new way of determining LCC and that the LCC-wildfire relationship is not the most robust would strengthen the manuscript.

We agree with the referee that adding more information about the robustness of the findings would increase the confidence on the results. Apart from calculating the risk of lightning-ignited wildfire from the variation of total lightning, LCC lightning and meteorological conditions, **we have alternatively calculated the risk by focusing on the ratio of LCC lightning to total lightning**, which does not depend on the lightning parameterization. We have obtained a global increase of lightning-ignited wildfire risk and a decrease in the pole by using both methods, which we think provides confidence on the results and, more importantly, clearly shows that parameterizing LCC lightning apart from total lightning is essential to investigate the sensitivity of lightning-ignited wildfires under climate change.

Additionally, I do have some issue that no CG lightning was utilized in this research. Relying on GLM, which cannot distinguish between IC and CG and have resolution and detection problems, could be a major sticking point for some readers and reviewers. I liked how the two approaches were explained, and I think that is sufficient for me. However, this point also supports my recommendation for couching your results and conclusions with more caveats and perspective, while strengthening your analysis with a few additional items and discussion.

We have now introduced the projected variation of CG lightning in the results. Interestingly, the obtained variation of CG lightning is more similar to the variation of LCC lightning than to the variation of total lightning, especially in northern latitudes. We think this result can provide more robustness to the conclusions by compensating the lack of distinction between CG and IC by GLM. Despite the introduction of CG lightning, we have not considered them to estimate the risk of lightning-ignited wildfires in Fig. 3 and 4 because their occurrence is based on a CG/IC ratio parameterization for the western states of the United States only (Price & Rind 1993).

Specific Comments

Lines 101-115: Why wasn't any cloud-to-ground lightning flash/stroke data used to supplement or augment this analysis? You cannot definitively declare that GLM detected lightning was IC or CG.

Lines 107-109: I don't think you can conclude anything about proportions of LCCs starting fires from these results. Not all LCCs are CG as well.

Answers for the previous two specific comments: We agree with the reviewer that we cannot establish what proportions of LCCs can start a fire by using only GLM lightning measurements. However, this is not in the scope of this paper. We have used GLM data to determine if LCC lightning have more chances to ignite a fire than total lightning without attempting to arrive at a precise estimate.

Although Fairman and Bitzer (2022) recognize that the LCC classification model can benefit from matching with ground-based lightning networks to classify CG and IC flashes, the method was trained to identify CG strokes with continuing current reported by HAMMA. Fairman and Bitzer (2022) state:

“it remains unclear if current flow in the final stage of IC flashes is of similar nature to continuing current processes in channels connecting to the ground. Due to no conclusive evidence of continuing current flow through the IC during the late stage of an IC flash, GLM groups associated with the late stage of an IC flash are not included in this analysis”.

Note that the method presented by Fairman and Bitzer (2022) improves the classification of lightning with CC with respect to Bitzer (2017) by removing the IC flashes with a pulsed leader emitting optical signal for more than 5 consecutive frames. Therefore, we consider that the GLM data and the classification method we have used is appropriate to determine whether LCC lightning tends to produce more fires than total lightning.

Line 119: Table 1, why 2091-92 and 2095 and not 2091-2095?

Due to storage limitations, we could not save the output for years 2093 and 2094.

Lines 146-149: Interesting result

Thanks.

Lines 152-185: This paragraph is a bit dense and tough to glean what the authors want to convey

Even agreeing in that, we think it is necessary to understand the simulated changes in the ratio of LCC to total lightning, which are closely related with the changes and the absolute values of the updraft mass flux.

Lines 163-166: Is this due to smaller sample size of thunderstorms and fewer variations on how and the types of thunderstorms develop?

The total number of thunderstorms in polar regions is significantly less than in other regions, which causes a smaller sample size and a larger standard deviation. **We have added this comment to the manuscript.**

Lines 166-170: Not quite understanding this statement. Why would an overall quadratic relationship lead to varying ratios, including perhaps regionally to hemisphere-wide? Trying to articulate changes or diverges in relationships between LCC and updraft flux?

We have removed this statement. We wanted to point out that the high variability of the upward mass flux in Northern Polar regions together with the quadratic relationship makes difficult visualizing how the ratio can response to climate change by simply showing the plots. We think this is clear without introducing this confusing statement.

Lines 172-175: What about high elevation thunderstorms where Totals Totals doesn't capture the convective potential well?

As the reviewer points out, the Total Totals index (i. e., the temperature and humidity vertical profiles) is not a good proxy for the development of thunderstorms in some areas, such as high elevation terrains. However, the purpose of this analysis is discussing the source of the change in the upward mass flux at a global/annual scale. The effect of high altitude terrains is included in the model.

Line 197: Maybe change dry to "drier"? "Dry" thunderstorms are somewhat regionally defined. If lightning to precipitation ratio changes, it doesn't necessarily mean it is "dry" as it may not reach those more regionally defined thresholds.

We agree. **We have changed "dry" to "drier".**

Lines 217 and 210-240: Change "will" to "may". This refers to my general comments where they are some assumptions stacking on assumptions here. Assuming that this method can't accurately model increases in LCC, then assuming that LCCs have a robust relationship with starting fires or a minimum more efficient than other CGs, then assuming a decrease in relative humidity would lead to an increase in lightning-ignited wildfires. I am not saying these are poor methods or the results are invalid, but I am cautioning the

authors to use caution when declaring relationships or conclusions based on this research without proper caveats or qualifiers.

Following these recommendations, **we have changed “will” to “may” and toned down the discussion and conclusions.** Apart from using changes in LCC lightning to estimate the risk of lightning-ignited wildfires, we have included changes in total lightning.

Figure 4: I would like to see more figures that show the potential changes in lightning-ignited wildfire patterns and frequencies. Are you able to produce a plot that shows where lightning and LCC frequency increases with lowered RH, higher temperatures, and possibly increased vapor pressure deficit? Using LCC with change in RH is a good start, but to improve the manuscript and better show how lightning-ignited wildfire patterns will change, I think another figure illustrating all those changes that lead to a likely increase in lightning-ignited wildfires would be helpful.

Done. **We have now showed a deeper analysis by including more variables,** as commented above.

Lines 288-290: Why is there a 66% increase? Could you go into this more?

This statement was not correct and **has been changed.**

Lines 307-312: I think there are other factors that are more important to lightning-ignited wildfires than LCC. I think changes in LCC should be a supplemental piece to the change, rather than the primary driver of your results and conclusions on lightning-ignited wildfire pattern changes.

We agree. **We have now included changes in total lightning, total precipitation, relative humidity, vapor pressure deficit and temperature as possible drivers of lightning-ignited wildfires** (apart from LCC lightning).

Lines 366-395: You do a good job of explaining some of the caveats and walking through your methods here. I think you just need to carry some of the caveats over to the results and discussion sections by changing wording (i.e., will to may, responsible to associated) and providing more context. •

We have toned down the discussion.

Lines 409-425: Why did you choose 2091, 2092, and 2095 and 2009-11 as your periods?

We chose 2009-2011 as representative present-day simulations. We chose the 2090s as the most extreme years of RCP6.0 scenario (RCP6.0 prescribed emissions are given until 2100). We did not include 2093 and 2094 for storage limitations, but initially we planned to include them in our study.

Reviewer #2 (Remarks to the Author):

This manuscript applies the first author's parameterization for long-continuing-current (LCC) lightning flashes to estimate the future change in LCC with global warming.

I do not trust the lightning parameterizations used here, so I do not have much faith in the results. The study parameterizes total lightning

as a function of cloud-top height over land using Price and Rind (1992) even though better schemes are available. Then, LCC is parameterized as a fraction of that total lightning, with the fraction given by a strange quadratic function published in the first author's 2022 paper in Geoscientific Model Development (GMD). I say "strange" because the LCC fraction is non-monotonic in the convective mass flux, coming back down to zero and then to negative values for sufficiently intense storms. The GMD paper gives no explanation or justification for this behavior. I tend to believe that the non-monotonic behavior is telling us that LCC should not be parameterized in terms of 2.5x2.5-degree mass flux. On the other hand, if the reported increase in fire is coming not from the quadratic dependence of LCC on mass flux, but from increases in the cloud-top height from the Price and Rind scheme for total lightning, then these results are not novel (see, e.g., Krause et al, 2004).

As the referee points out, we use a non-monotonic relationship between the 2.5x2.5-degree mass flux and the ratio of LCC lightning to total lightning. However, we do not agree that a non-monotonic relationship means that the LCC occurrence should not be parameterized in terms of the 2.5x2.5-degree mass flux.

On the one hand, non-monotonic relationships between meteorological variables and atmospheric electricity are commonly found in literature [Takahashi (1978), Saunders (1993)]. The vertical profiles of the temperature, moisture and aerosol concentration can non-linearly influence the electrical charging of hydrometeors. The concentration of aerosols and the mass flux can strongly influence the formation and size of cloud droplets, determining the rate of collision and coalescence of raindrops [Nakajima et al., (2001); Tao et al., (2012)]. For example, it is accepted that there is a non-linear relationship between the concentration of Cloud Condensation Nuclei (CCN) influencing the microphysics in thunderstorms and lightning activity [e. g., Mansell and Ziegler (2013)]. The humidity and temperature where collision of hydrometeors occur determine the charging of graupel and ice, strongly influencing the structure of the electrical charged layers in thunderstorms [Takahashi, (1978)].

In the case of LCC lightning, Bitzer (2017) reported for the first time a non-homogeneous seasonal and spatial ratio of LCC lightning to total lightning. This result was reproduced by Pérez-Invernón et al. (2021) and by Fairman and Bitzer (2022). The non-homogeneous ratio of LCC lightning to total lightning suggest that it depends on meteorological parameters. Bitzer (2017) noted that the ratio tends to be lower during winter and in the oceans, where thunderstorms tend to have weaker mass flux. In our GMD 2021 paper, we determined that there is a possible quadratic relationship between the mass flux and the ratio of LCC to total lightning. We showed that the reported relationship can reproduce fairly well the ratio of LCC to total lightning reported by LIS onboard the ISS. We can also compare the simulated ratio of LCC to total lightning (GMD paper) with the recently ratio reported by GLM (Fairman and Bitzer), obtaining a good agreement at latitudes where LIS did not provide measurements.

At this point, one can ask if a non-monotonic relationship between LCC lightning and the mass flux is reasonable. Firstly, it is important to explain the origin of the continuing current in lightning. Lapierre et al. (2017) reported that in-cloud negative leaders of positive cloud-to-ground lightning can inject high amounts of charges in the return stroke, producing a long continuing current. However, Lapierre et al. (2014) did not find any clear relationship between the propagation of in-cloud positive leaders and negative cloud-to-ground lightning. Their results suggest that the structure of electrical charges in thunderclouds could have a double effect in the occurrence of continuing currents: 1) On the one hand, the structure of the electrical charged layers plays an

important role in the polarity of cloud-to-ground lightning [e. g., Rust et al. (2005)]. In the same manner, the structure of electrical charged layers in thunderstorms is closely related with the meteorological conditions. According to Lapierre et al. (2014 & 2017), the physical mechanisms producing continuing current in negative and positive cloud-to-ground lightning are different. Therefore, we can expect a non-linear relationship between the meteorological conditions of thunderstorms and continuing currents. 2) On the other hand, Lapierre et al. (2017) reported that the propagation of the in-cloud negative leader supplies the continuing current of positive cloud-to-ground lightning. The propagation of the negative leader is influenced by the distribution of electrical charges in thunderstorm, introducing complexity to the mechanism behind the occurrence of the continuing current. **We have introduced this explanation in the manuscript.**

Given the complexity and uncertainties in the formation of continuing current in lightning and the apparent relationships with the structure of electrical charged layers in thunderstorm, we think that obtaining a non-monotonic relationship between the ratio of LCC and total lightning is completely reasonable. More investigation is needed to understand the relationships between continuing currents in negative and positive cloud-to-ground lightning and the meteorological conditions of thunderstorms. However, such investigations are not available yet and we can only propose a parameterization of LCC lightning and compare the simulated and the observed climatologies in order to test its validity.

References (now introduced in the manuscript)

- Takahashi, Tsutomu. "Riming electrification as a charge generation mechanism in thunderstorms." *Journal of Atmospheric Sciences* 35.8 (1978): 1536-1548.
- Saunders, C. P. R. "A review of thunderstorm electrification processes." *Journal of Applied Meteorology and Climatology* 32.4 (1993): 642-655.
- Nakajima, Teruyuki, et al. "A possible correlation between satellite-derived cloud and aerosol microphysical parameters." *Geophysical Research Letters* 28.7 (2001): 1171-1174.
- Rust, W. David, et al. "Inverted-polarity electrical structures in thunderstorms in the Severe Thunderstorm Electrification and Precipitation Study (STEPS)." *Atmospheric Research* 76.1-4 (2005): 247-271.
- Tao, Wei-Kuo, et al. "Impact of aerosols on convective clouds and precipitation." *Reviews of Geophysics* 50.2 (2012).
- Mansell, Edward R., and Conrad L. Ziegler. "Aerosol effects on simulated storm electrification and precipitation in a two-moment bulk microphysics model." *Journal of the Atmospheric Sciences* 70.7 (2013): 2032-2050.
- Lapierre, Jeff L., et al. "On the relationship between continuing current and positive leader growth." *Journal of Geophysical Research: Atmospheres* 119.22 (2014): 12-479.
- Lapierre, Jeff L., et al. "Expanding on the relationship between continuing current and in-cloud leader growth." *Journal of Geophysical Research: Atmospheres* 122.8 (2017): 4150-4164.
- Pérez-Invernón, Francisco J., et al. "Influence of the COVID-19 lockdown on lightning activity in the Po Valley." *Atmospheric Research* 263 (2021): 105808.

Reviewer #3 (Remarks to the Author):

In this study, the authors begin by arguing using observations that long continuous current (LCC) lightning could account for 12 to 68% of lightning-ignited fires in the United States. They argue that since LCC flashes account for less than 10% of total lightning flashes, this suggests that LCC flashes are more effective than regular flashes at starting fires, meaning that any changes in LCC lightning in particular

might be something we want to pay close attention to under climate change. To investigate that further, they then apply a recently developed parameterization for the fraction of LCC lightning, based on updraft mass flux, in simulations in present-day and future climates using the ECHAM5 model. For a sense of the total flash rate over land in each simulation, a traditional cloud-top-height-based parameterization is used.

Their primary finding is that fire risk may decrease with warming in the Boreal regions due to an increase in precipitation and little-to-no significant change in the LCC lightning flash density. This is something that would not have been possible to determine by looking at changes in total lightning flash density alone, and runs in contrast to the findings of Krause et al. (2014). In other regions, LCC lightning flash density tends to increase, and this is often accompanied by an increase in precipitation making changes in overall lightning-ignited fire risk less clear. There is understandably a fair amount of uncertainty associated with these results, however, which I think the authors do a reasonable job of communicating, but there are places where I think they could do better, which I cite in my major comments.

In general I agree with their conclusion that the contrasting results with Krause et al. (2014), who used a more comprehensive setup with an atmosphere model coupled to a land surface model that included fires, motivate a followup study in which LCC lightning, and in particular the connection between LCC lightning and fire ignition, are included in a similar Earth system modeling framework. While the authors work hard to approximately capture changes in lightning-ignited fire risk using physically motivated logic, these estimates would be more robust with a model that helped explicitly estimate changes in area burned. More years of simulation in the present-day and future climates would also contribute to increased confidence in the conclusion. Overall though, with some improvements, the publication of this manuscript could serve as a good precursor for that followup work.

We thank the reviewer for these encouraging comments. As the reviewer points out, there are different ways to improve the present study in the future. This study could be extended in the future with more data on LCC lightning reported by GLM over America and by MTG-LI over Europe and Africa. In addition, coupling the LCC parameterization with a vegetation module could also give an estimation of the area burned. However, we think that this study is a breakthrough because it lays the groundwork for using parameterizations of LCC lightning to investigate the sensitivity of lightning-ignited wildfires under climate change. Although the burned area is not computed, the analysis of the variation of total lightning, LCC lightning and meteorological conditions provide enough data to determine in which regions the burned area would increase or decrease.

Major comments

- While for some area-averaged regional change metrics a t-test is used to assess statistical significance (e.g. in the first paragraph of Section 2.2), no test appears to be applied to changes in individual gridpoints on maps. I think it would make the results of this study easier to interpret if a similar t-test were applied to individual

gridpoints, and then a mask (or hatching) were applied to all gridpoints where the change was not statistically significant in the maps. I believe this would be useful since in the text a number of important conclusions are made regarding specific (fairly local) regions, and it would be good to be able to confirm whether these changes are statistically significant. Something similar could be said for the scatterplots in Figure 4. Some gridpoint-level changes in LCC lightning flash density in particular are quite small and seem they may not be outside the range of interannual variability. Should they be included in the percent of points with decreased or increased LIW risk?

Following this suggestion, **we have now applied a T-test to all the variables plotted in the manuscript to evaluate the risk of lightning-ignited wildfire** (total lightning, LCC lightning, total rain, relative humidity, vapor pressure deficit and temperature). As explained in the manuscript, we have only considered that the changes are significant if the p-value is below 0.05. Then, we have masked all the grid cells where the changes are not statistically significant. In addition, **we have removed the scatterplots in Figure 4 and replaced them by a map showing the spatial distribution of the variation of the risk of lightning-ignited wildfire**. We think that the new figure provides a clearer view of the points where the changes are large or small.

- Are there other possible choices for an LCC fraction parameterization? Could that introduce uncertainty? How much do we trust the existing scheme, whose details--importantly the threshold at which the LCC ratio begins to decrease with increasing updraft mass flux--may depend on the circulation model that was used to generate the updraft convective mass fluxes to fit it?

For the discussion on the uncertainty in the parameterization of the LCC lightning, we refer to the answer provided to Reviewer 2.

Regarding the scale factor applied to the updraft mass flux, we explained how it is calculated for MESSy in our GMD paper. As the reviewer points out, this scale factor may depend on the model. However, it is out of the scope of this paper determining how it could vary between different models, as we use the same model as in our previous paper (MESSy). Regarding the threshold at which the LCC ratio begins to decrease with increasing updraft mass flux, we agree with the reviewer that it could depend on the used model. However, as we showed in our previous paper, the parameterization used produces a good agreement between the observed and the simulated LCC lightning distribution by using MESSy.

Is the uniform scale factor applied to the updraft mass fluxes before passing them into the scheme enough to address this? Since the LCC scheme is fairly central to the conclusions of this study, including some discussion of this feels important.

Yes, the uniform scale factor is applied to the updraft mass fluxes before passing them into the scheme. **We have now pointed out that the scale factor for the updraft mass flux is applied “online”**. This scheme produces a good agreement between simulations and observations when using MESSy (GMD paper).

Specific comments

Lines 48-53: were these studies all using the same emissions scenarios? That seems like something important to control for; otherwise it might

be better to report % increase or decrease per degree warming.

No, they were not. Some studies chose the RCP8.0 scenario. **We have now included the % increase/decrease per degree in Table 1.**

Line 64: "atmospheric model European Center HAMburg general circulation model 6 (ECHAM6)" having "atmospheric model" and "general circulation model" in such close succession is a bit awkward to read and feels somewhat redundant. Could one of those descriptors be dropped or are they part of the official name?

We have removed "atmospheric model".

Lines 79-82: does focusing on the relationship between LCC lightning and fires only in the United States bias these results at all? Could looking at different land and fire regions (e.g. with different vegetation types) change these conclusions?

There is a considerable total number of ecoregions in the United States that are also present in other parts of the world (source: Commission for Environmental Cooperation). Therefore, we can assume that focusing only in the United States does not introduce a significant level of uncertainty in our results. In the future, we will expand this analysis by using LCC lightning data over Europe, where there is enough fire data from local/national institutions (see Pérez-Invernón et al. (2021, ACP)).

Lines 101-115: it is understandable that measuring the causal relationship between lightning and fires is difficult from satellite data alone. Thank you for acknowledging the large uncertainty in these statistics.

Thank you.

Lines 131-134: I just want to confirm that this is over land as well (if it is, maybe refer to this as "polar land regions"). If not, it seems that it would be more relevant if it were focused on land, since it is used to comment on the risk of lightning-ignited fires in polar forests. More generally, while I think it is fair to include the global total lightning change statistics to compare to previous literature, the focus of this paper should be over land, since that is where fires occur.

Thank you. It was not only over land. **We have now focused on data over land.**

Table 1: for completeness it might be good to include the total land lightning flash rate statistics (to compare with the LCC land lightning statistics).

Done.

Lines 137-139: I suppose since some fires are started by non-LCC lightning that an increase in total lightning may still result in an increase in lightning-ignited fires (even if LCC lightning is

effectively constant). Maybe phrase this as "[...] may not be as influenced by the future increase of lightning activity as suggested in previous studies."

Done.

Lines 142-145: to my eye, particularly over land the spatial pattern of the change in total and LCC lightning looks to be quite similar to first order. There are in indeed exceptions, but they are difficult to pick out. Perhaps adding a panel that showed a map of where the change was statistically significant in both the total and LCC cases, but of opposite sign would help the reader confirm what is written in the text a little more quickly.

We have now added maps where the changes are statistically significant (see Figure S1). In addition, Figure 2(a) shows the variation of the ratio of LCC to total lightning.

Lines 145-149: in looking at the figure in most of these regions it looks more like total lightning decreases and LCC lightning increases. Is this statement correct?

Thank you. This statement was not correct and has been corrected.

Lines 158-159: how is it determined what a "thunderstorm" is in this context? Is it whether the total lightning parameterization predicts non-zero lightning?

Figure 2: the colormaps in (a) and (b) are very difficult to read near zero -- they seem to perhaps make small changes appear larger than they really are. Is there a way to address this?

We have changed the colorbar so that values near zero are easier to read.

Figure 2(a-e): I recommend using symmetric colorbars centered at zero for these change plots. It generally makes them easier to read and interpret.

Done.

Figure 2(a-e): this is a bit of a style nit-pick, but more generally, why are the colorbar tick labels unevenly spaced with different formats? With the exception of panel (f), I think there is no need to use scientific notation (e.g. " 5.7×10^{-1} " is more clearly represented as "0.57"; " 1.5×10^1 " is more clearly represented as "15" etc.).

We have now used the same format for all the panels.

Lines 192-193: does "dry" lightning ever occur in these simulations? In that case this ratio would be unboundedly large. Also, why restrict to using the convective rain rate in the denominator? Would total rain rate not make more sense here?

We have now used total rain instead of only the convective rain rate. In addition, **we have changed “dry” by “drier”**, as we are not able to define what can be considered as “dry lightning” in each region of the world.

Line 205: typo "Southeastern"

Changed.

Figure 3: the colorbar is saturated in most regions of the world here. I would suggest extending the range. Perhaps more generally though, would it make more sense to pose this as a percent change? I don't have a very good handle on the order of magnitude of this ratio in the absolute sense, and I suspect other readers do not either, so it is hard to determine whether these changes are substantial deviations from the climatology or not.

We have changed this figure by showing the % change. We consider it is clearer now.

Line 218: does "the globe" include or exclude the polar regions? Would it be more effective to use non-overlapping regions?

Here, “the globe” includes polar regions. We think providing the results for the entire globe could be more useful for the readers, and differences with polar regions can already be seen by following this approach.

Figure 4: can you confirm whether these changes are considered only over land? I assume they are, but I just want to make sure.

We have now ensured that changes are considered only over land.

Lines 419-420: is there a reason for excluding years 2093 and 2094 from the analysis? Is it just that the output did not exist?

Due to storage limitations, the output does not exist.

Line 459: where does the factor of 6.57 come from? I think you mention it briefly in the caption of Figure 2, but a more detailed explanation could be helpful (i.e. acknowledging that it is needed to scale the the updraft mass fluxes in ECHAM such that they approximately match the magnitude of the updraft mass fluxes in ERA5, the dataset the parameterization was derived with).

We refer to ur GMD paper (Pérez-Invernón et al. (2021, GMD)) for more details on the origin of the factor of 6.57.

REVIEWER COMMENTS

Reviewer #2 (Remarks to the Author):

The discussion added to the manuscript has not altered my low confidence in the parameterizations being used in this modeling exercise. A forecast is only as good as the model it is based on, and I do not see the merit in this one.

Reviewer #3 (Remarks to the Author):

I thank the authors for working to address the reviewers' comments. I believe they have addressed most of them in the revised manuscript. However, I have a couple remaining concerns, which I elaborate on more in my specific comments. These relate to the framing of the discussion of the spatial patterns of change in total and LCC lightning, and the new method used to estimate the change in fire risk between the present and future climate.

General comments

I'm not sure about the style guidelines of Nature Communications, but I know in other journals it is common practice to refer to supplemental figures from within the manuscript so that the reader is aware of what other context is available for the work. Would it be possible for you to describe and reference the key features of those figures in the main text where relevant?

Specific comments (line numbers refer to the tracked-changes manuscript)

Line 24: mention that this percent increase is global.

Line 25: I suggest using "North" America to be consistent with the usage earlier in the sentence for "South America."

Line 134: just confirming that the change in total lightning did not change when considering only polar land regions?

Lines 132-143: could you also report the p-value for the change in total lightning over the next century in the polar regions, as is done for LCC and CG lightning?

Lines 144-164: in looking at Figure 1 in the originally submitted manuscript and Figure 1 in the revised manuscript, the panels representing the change in total lightning and the change in LCC lightning do not look identical (to the point that in some regions it looks as though the plotted changes change sign, e.g. the west coast of Canada in both panels, or southern South America in the total lightning panel). Why is that? Is it just a plotting artifact?

Lines 144-164 (continued): If we look at Figure S4--where changes are plotted as percentages and statistical significance is taken into account--the panels for total lightning and LCC lightning look qualitatively the same (essentially the same sign changes in the same locations). To me this underscores the fact that predicted changes in total lightning and LCC lightning more often than not have the same sign, and so I find the statement "However, the change of the global distribution of CG and LCC lightning flashes is not similar (Fig. 1, b, c)" to be a bit exaggerated, so I suggest rephrasing it. It seems like the patterns of change are similar, and Figure S4 suggests there are few places where the two have statistically significant differently signed changes. It might be better to convey that while in most locations the sign of the change in LCC and total lightning is similar, there are locations where it may not be (noting where those are), and more years of simulation might be required to make those findings more robust.

Figure S4: in addition to being masked for statistical significance, are these also masked over ocean?

If not, I'm surprised that there are so few statistically significant changes over the oceans. For instance, based on Figure 1(a) I would have guessed that changes in total lightning over the tropical oceans would be significant in many places. If there is also masking over ocean, I might recommend removing that, since for maps (in contrast to regional averages) it does not take anything away to include values over ocean where they are significant.

Table 1 and elsewhere in the manuscript: when referring to polar land, are the authors referring to just the north pole region, or also the south pole region? My sense is that it is just the north pole region (it would not make sense to include the south pole region since nearly all land there is ice covered, without wildfire fuel). If that is correct, please clarify that in the manuscript.

Lines 176-179: how is it determined what a "thunderstorm" is in this context? Is it whether the total lightning parameterization predicts non-zero lightning? I asked this question in my previous review, but it was not addressed. Is this merely looking at the updraft mass flux during all times (not just thunderstorms)?

Line 209: I believe this is now the ratio of LCC lightning to "total rain."

Lines 215-216: "We estimate a decreasing risk of lightning-ignited wildfires in polar regions in the 2090s [...]" I believe this is referring to results in Figure 3; however due to masking due to statistical significance this is now less apparent. Please double check that the results in this paragraph are consistent with what is shown in the figure.

Lines 232-238: I appreciate the authors looking into doing something different than the old Figure 4 to approximately quantify the change in fire risk over the next century. However, while I see where this is coming from, it feels a bit ad hoc. Is there reason to expect that the percentage changes of all of these factors would add linearly to produce an index of fire risk change? Intuitively it seems like some factors might be more important than others, so unless all statistically significant factor changes point in the same fire risk change direction, it may be difficult to determine which win out at a particular location.

Lines 232-238 (continued): I wonder if there is anything grounded in previous literature that could be done with the available diagnostics. For instance Steinfeld et al. (2022) reviews a number of "fire weather indices" which might be applicable to quantify this risk in each climate (and by extension the change). In addition, I think the location of wildfire fuel should be taken into account in the analysis--if not quantitatively, at least qualitatively. This stands out most to me in the discussion of changes in risk in the polar regions (see my next comment), though something similar could be said about changes over the Sahara Desert. I do not know the full set and time frequency of diagnostics you have available from these runs, however, and so I understand this may make doing a more advanced assessment of changes in fire risk more challenging.

Lines 240-244: by this metric it appears the largest positive change in the north pole region is over the ice-covered area of Greenland, which is not a region where I would expect wildfires to occur due to fuel availability constraints. Over the Boreal regions, where there is available wildfire fuel, there appears to generally be a negative change. It feels as though this could also be taken into account in this discussion.

Lines 534-535: "These results suggest that the structure of electrical charges in thunderstorms could have a double effect [...]" The phrase "a double effect" seems to imply a sort of quantification; however my reading of the subsequent discussion does not seem to suggest this. Perhaps I would consider replacing "a double effect" with "multiple effects."

References

Steinfeld, D., Peter, A., Martius, O., & Brönnimann, S. (2022). Assessing the performance of various

fire weather indices for wildfire occurrence in Northern Switzerland. EGU sphere, 1–23.
<https://doi.org/10.5194/egusphere-2022-92>.

Response to reviewers

We appreciate the time and effort that the reviewers dedicated to providing feedback on our manuscript and are grateful for the insightful comments and valuable improvements to our paper. In this second revision, we have incorporated most of the suggestions made by the Reviewer #3. Those changes are highlighted within the manuscript. Please see below, in blue, for a point-by-point response to the reviewers' comments and concerns.

Reviewer #2 (Remarks to the Author):

The discussion added to the manuscript has not altered my low confidence in the parameterizations being used in this modeling exercise. A forecast is only as good as the model it is based on, and I do not see the merit in this one.

As we showed in Pérez-Invernón et al. (2021, <https://doi.org/10.5194/gmd-15-1545-2022>), the accuracy of the implemented parameterization of LCC lightning is limited but can produce a reasonable agreement with space-based lightning measurements provided by TRMM-LIS and ISS-LIS. In turn, we have recently presented a comparison of our simulations with the recent climatology of LCC lightning provided by the Geostationary Lightning Mapper (GLM) over the Americas (see Perez-Invernón, AGU2023). This new comparison indicates a good agreement between the simulated ratio of LCC lightning to total lightning and continuous lightning measurements from a geostationary orbit:

Figure 1: Ratio of LCC to total lightning based on lightning measurements provided by GLM during 2019

It is also important to note that variations in the occurrence of LCC lightning is not the only proxy for lightning-ignited wildfire risk estimation in this work. We have added an analysis of changes of several meteorological factors (total lightning, temperature, precipitation, relative humidity, vapor pressure deficit).

Therefore, we leave the editor to judge the confidence of the modeling results presented in this work.

References

Perez-Invernon, F. J., Huntrieser, H., Joeckel, P., & Gordillo-Vazquez, F. J. Parameterization of long-continuing-current lightning for chemistry-climate models. In Fall Meeting 2022. AGU. agu2022fallmeeting-agu.ipostersessions.com/Default.aspx?s=12-E7-C9-4E-2A-EC-35-6C-6D-EC-57-E8-4B-13-AF-14

Reviewer #3 (Remarks to the Author):

I thank the authors for working to address the reviewers' comments. I believe they have addressed most of them in the revised manuscript. However, I have a couple remaining concerns, which I elaborate on more in my specific comments. These relate to the framing of the discussion of the spatial patterns of change in total and LCC lightning, and the new method used to estimate the change in fire risk between the present and future climate.

General comments

I'm not sure about the style guidelines of Nature Communications, but I know in other journals it is common practice to refer to supplemental figures from within the manuscript so that the reader is aware of what other context is available for the work. Would it be possible for you to describe and reference the key features of those figures in the main text where relevant?

We thank the reviewer for these encouraging comments.

Please note that all the figures included in the supplement are also cited in the manuscript.

Specific comments (line numbers refer to the tracked-changes manuscript)

Line 24: mention that this percent increase is global.

Done.

Line 25: I suggest using "North" America to be consistent with the usage earlier in the sentence for "South America."

Done.

Line 134: just confirming that the change in total lightning did not change when considering only polar land regions?

Total lightning did change when considering only polar regions (~56%), but the change in the occurrence of LCC lightning (~21%) was significantly lower than the change in total lightning.

Lines 132-143: could you also report the p-value for the change in total lightning over the next century in the polar regions, as is done for LCC and CG lightning?

We have included the p-value for the change in total lightning in polar-land regions:

“The calculated p-value for the projected and the present-day LCC lightning rates in land polar regions is 0.38, while the p-value for the projected and the present-day total and CG lightning rates in land polar regions are 0.32 and 0.12, respectively”.

Lines 144-164: in looking at Figure 1 in the originally submitted manuscript and Figure 1 in the revised manuscript, the panels representing the change in total lightning and the change in LCC lightning do not look identical (to the point that in some regions it looks as though the plotted changes change sign, e.g. the west coast of Canada in both panels, or southern South America in the total lightning panel). Why is that? Is it just a plotting artifact?

As the reviewer points out, we have changed the colorbar in Figure 1. The colorbar in Figure 1 in the originally submitted manuscript was less sensitive to small values and the white color was not correctly assigned to zero, causing that the blue color was assigned to small positive values. This was solved by changing the scale in the colorbar.

Lines 144-164 (continued): If we look at Figure S4--where changes are plotted as percentages and statistical significance is taken into account--the panels for total lightning and LCC lightning look qualitatively the same (essentially the same sign changes in the same locations). To me this underscores the fact that predicted changes in total lightning and LCC lightning more often than not have the same sign, and so I find the statement "However, the change of the global distribution of CG and LCC lightning flashes is not similar (Fig. 1, b, c)" to be a bit exaggerated, so I suggest rephrasing it. It seems like the patterns of change are similar, and Figure S4 suggests there are few places where the two have statistically significant differently signed changes. It might be better to convey that while in most locations the sign of the change in LCC and total lightning is similar, there are locations where it may not be (noting where those are), and more years of simulation might be required to make those findings more robust.

We have rephrased:

“While in most locations the sign of the change in LCC and CG lightning is similar, there are locations where it may not be”

Figure S4: in addition to being masked for statistical significance, are these also masked over ocean? If not, I'm surprised that there are so few statistically significant changes over the oceans. For instance, based on Figure 1(a) I would have guessed that changes in total lightning over the tropical oceans would be significant in many places. If there is also masking over ocean, I might recommend removing that, since for maps (in contrast to regional averages) it does not take anything away to include values over ocean where they are significant.

Yes, Figure S4 is marked over ocean. We have now mentioned this in the caption. We have masked over oceans because this plot represents the risk of lightning-ignited wildfires, that is zero over ocean. We think that including the ocean would introduce confusion to the reader.

Table 1 and elsewhere in the manuscript: when referring to polar land, are the authors referring to just the north pole region, or also the south pole region? My sense is that it is just the north pole region (it would not make sense to include the south pole region since nearly all land there is ice covered, without wildfire fuel). If that is correct, please clarify that in the manuscript.

We are referring to North Pole, since we focus on lightning-ignited wildfires (not present in the South Pole). We have now clarified this in the caption of Table I.

Lines 176-179: how is it determined what a "thunderstorm" is in this context? Is it whether the total lightning parameterization predicts non-zero lightning? I asked this question in my previous review, but it was not addressed. Is this merely looking at the updraft mass flux during all times (not just thunderstorms)?

We added the definition of “thunderstorm” in the simulation in the caption of Figure 2. We have now added it to the main text:

“The criterion to consider the MUMF in the calculation of the annual average is that the cloud thickness is at least 3 km and that the lightning occurrence rate is larger than $1.433 \times 10^{-14} \text{ m}^{-2}\text{s}^{-1}$) [Tost et al. (2007)]”

Line 209: I believe this is now the ratio of LCC lightning to "total rain."

This is now the ratio of LCC lightning to total rain rate (total rate during 10 h).

Lines 215-216: "We estimate a decreasing risk of lightning-ignited wildfires in polar regions in the 2090s [...]" I believe this is referring to results in Figure 3; however due to masking due to statistical significance this is now less apparent. Please double check that the results in this paragraph are consistent with what is shown in the figure.

Yes, we refer to Figure 3. We have checked that the results in the text are consistent with Figure 3.

Lines 232-238: I appreciate the authors looking into doing something different than the old Figure 4 to approximately quantify the change in fire risk over the next century. However, while I see where this is coming from, it feels a bit ad hoc. Is there reason to expect that the percentage changes of all of these factors would add linearly to produce an index of fire risk change? Intuitively it seems like some factors might be more important than others, so unless all statistically significant factor changes point in the same fire risk change direction, it may be difficult to determine which win out at a particular location.

Lines 232-238 (continued): I wonder if there is anything grounded in previous literature that could be done with the available diagnostics. For instance Steinfeld et al. (2022) reviews a number of "fire weather indices" which might be applicable to quantify this risk in each climate (and by extension the change). In addition, I think the location of wildfire fuel should be taken into account in the analysis--if not quantitatively, at least qualitatively. This stands out most to me in the discussion of changes in risk in the polar regions (see my next comment), though something similar could be said about changes over the Sahara Desert. I do not know the full set and time frequency of diagnostics you have available from these runs, however, and so I understand this may make doing a more advanced assessment of changes in fire risk more challenging.

As the reviewer points out, Figure 4 shows an estimation of the variation in the risk of lightning-ignited wildfire without considering nonlinearity in the included factors. We agree that changes in the factors would not add linearly to produce fire risk change. However, we think our analysis can provide useful information about changes in the risk of ignition under climate change.

Fire weather indices are combine meteorological variables to estimate the risk of wildfire. According to literature, there exists a wide range of fire weather indices combining meteorological parameters in a non-linear way. However, the particularities of each region mean that the application of one or the other index is more convenient. For example, the paper cited by the reviewer (Steinfeld et al. (2022)) shows that the applicability of the index in Switzerland depends on its mountainous terrain. Vant-Hull et al. (2018), Pérez-Invernón et al. (2021) and Pérez-Invernón et al. (2022) show that the precipitation threshold for the ignition of wildfires by lightning is strongly influenced by the region, etc. As a consequence, establishing a global index for the risk of lightning-ignited wildfire is not affordable today. Therefore, in this work we analyze the variation of the main meteorological parameters usually correlated with the risk of ignition. We do not intend to provide a quantitative estimation of the variation of lightning-ignited wildfire risk at a regional

scale, but an analysis of future changes at a global scale by using a chemistry-climate model. We think our results can motivate future analysis of the risk of lightning-ignited wildfire at a regional scale by using mesoscale atmospheric models, local fire weather indices and a parameterization of LCC lightning.

We have added a more regional discussion to Figure 3:

“Increases in northern polar regions are mainly focused in Greenland, where there is a low fuel availability. On the contrary, lightning-ignited wildfire risk decreases in northern polar regions are distributed across Siberia and Alaska, two regions where lightning-ignited wildfires are common during the fire season. Therefore, the simulations show a possible tendency to a lower risk of lightning-ignited wildfires in northern polar forest under climate change due to an increase in precipitation in combination with negligible variations in the occurrence of total and LCC lightning.”

Lines 240-244: by this metric it appears the largest positive change in the north pole region is over the ice-covered area of Greenland, which is not a region where I would expect wildfires to occur due to fuel availability constraints. Over the Boreal regions, where there is available wildfire fuel, there appears to generally be a negative change. It feels as though this could also be taken into account in this discussion.

We have now included the following paragraph in the discussion:

“In addition, we have obtained that the ratio of LCC lightning to total lightning will decrease in northern polar regions by the end of the century. In particular, we have found a significant decrease in the ratio of LCC lightning to total lightning over northern polar forest, while the enhancements in polar regions are located in Greenland, which is a region mainly covered by ice without fuel availability for wildfires.”

Lines 534-535: "These results suggest that the structure of electrical charges in thunderstorms could have a double effect [...]" The phrase "a double effect" seems to imply a sort of quantification; however my reading of the subsequent discussion does not seem to suggest this. Perhaps I would consider replacing "a double effect" with "multiple effects."

We have now replaced “double effect” by “multiple effects”.

References

Steinfeld, D., Peter, A., Martius, O., & Brönnimann, S. (2022). Assessing the performance of various fire weather indices for wildfire occurrence in Northern Switzerland. *EGUsphere*, 1–23. <https://doi.org/10.5194/egusphere-2022-92>.

Response to reviewers

Reviewer comment:

In the response letter you mention that the ocean mask is now mentioned in the caption of Figure S4, but I don't see that, please take a look and add this if it's missing.

We have added the information about the ocean mask to the caption of Figure S4.